# TOPOLOGY MATTERS IN RTL CIRCUIT REPRESENTATION LEARNING

**Mingyu Zhao**[1,*] **Xun He**[1,*]**, Jiawei Liu**[2]**, Jianwang Zhai**[1,†]**, Chuan Shi**[1,†]
[1]Beijing University of Posts and Telecommunications, [2]The Chinese University of Hong Kong
`zhaomingyu@bupt.edu.cn, hexun@bupt.edu.cn, liujw@cse.cuhk.edu.hk`
`zhaijw@bupt.edu.cn, shichuan@bupt.edu.cn`

## ABSTRACT

Representation learning for register transfer level (RTL) circuits is fundamental to enabling accurate performance, power, and area (PPA) prediction, efficient circuit generation, and retrieval in automated chip design. Unlike general programming languages, RTL is inherently a structured dataflow graph where semantics are intrinsically bound to the topology from a hardware view. However, existing language-model-based approaches ignore the nature of RTL circuits and fail to capture topology-sensitive properties, leading to incomplete representation and limited performance for diverse downstream tasks. To address this, we introduce TopoRTL, a novel framework that explicitly learns topological differences across RTL circuits and preserves the behavior information. First, we decompose RTL designs into register cones and construct dual modalities initialized with behavior-aware tokenizers. Second, we design three topology-aware positional encodings and leverage attention mechanisms to enable the model to distinguish topological variations among register cones and RTL designs. Finally, we introduce a topology-guided cross-modal alignment strategy, employing contrastive learning over interleaved modality pairs under topological constraints to enforce semantic consistency and achieve superior modality alignment. Experiments demonstrate that explicit topological modeling is critical to improving RTL representation quality, and TopoRTL significantly outperforms existing methods across multiple downstream tasks.

## 1 INTRODUCTION

Artificial intelligence is transforming electronic design automation (EDA) through representation learning. This approach maps circuits across abstraction levels into low-dimensional vector spaces, enabling unified modeling for critical tasks like PPA prediction, SAT solving, and circuit generation (Li et al., 2022b; Shi et al., 2023; 2024; Zheng et al., 2025; Liu et al., 2024b; 2025a;b; Fang et al., 2025). This capability supports the design left-shift paradigm, moving performance prediction and issue detection to earlier stages, which reduces costs and accelerates optimization (Xing, 2024; Zeng, 2024).

Among digital circuit abstractions, register-transfer level (RTL) is crucial. It is typically described using Verilog as the industry-standard hardware description language. Naturally, many approaches treat RTL as software programming code, focusing on learning syntax and semantic meaning through text-based representations. For example, CodeV (Zhao et al., 2025) uses GPT-3.5 to generate natural language descriptions from high-quality Verilog code and fine-tunes different large language models (LLMs) to enhance Verilog generation. Similarly, DeepRTL (Liu et al., 2025a) fine-tunes CodeT5+ on datasets connecting Verilog code to detailed descriptions, excelling in understanding and generating RTL. DeepRTL2 (Liu et al., 2025b) further integrates generation and embedding tasks within a unified framework.

Unlike software programming languages, RTL is **inherently a structured dataflow graph where behavior and topology coexist from a hardware view**. It explicitly specifies the flow of data

---

*Equal contribution. †Corresponding authors.

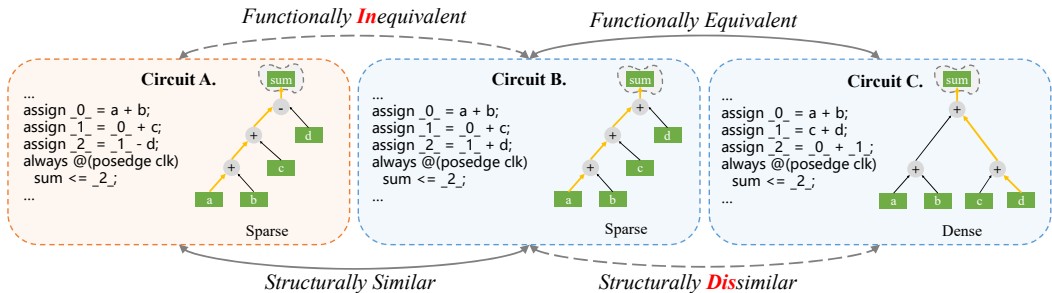

Figure 1: RTL is a structured dataflow paradigm where behavioral intent is inseparable from dataflow topology. Circuit A and B share a similar topology but implement different functions. Circuit B and C implement identical four-input adders but with divergent topologies.

between hardware registers and the logical operations performed on that data, which reflects quite closely the logic structure of the circuit being modeled (IEEE, 2006). Crucially, RTL is not a purely behavioral description (which abstracts away hardware structure) nor a purely structural one (which specifies gate-level connectivity). Instead, it represents a structured dataflow paradigm where behavioral intent is inseparable from topology (Micheli, 1994). This tight coupling between behavior and topology necessitates that RTL not be treated as a general programming language to learn.

While this text-based approach seems straightforward, we argue that *topology matters in RTL representation learning*. The topological structure of circuits directly influences their physical constraints and implementation details (Micheli, 1994). For instance, in Figure 1, Circuits A and B have similar topologies but produce different functions, while Circuits B and C, both four-input adders, demonstrate performance variations due to their topological differences. Circuit B's chain structure is less timely but more power-efficient than Circuit C's tree structure. Current methods typically use text-based approaches, often relying on LLMs that struggle with graph-structured data (Li et al., 2024), making it challenging to capture circuit topological properties, leading to the following question:

> *Can we model RTL circuits by incorporating*
> *both behavioral functions and topological structure information?*

To address the question, we analyze the fundamental nature of RTL circuits. As previously mentioned, the sequential RTL circuit consists of registers and combinational logic. When a signal propagates through the circuit, it undergoes a cyclic process:

**Computation Phase**. Signals traverse through combinational logic networks where functional transformations occur. This phase determines the circuit's operational behavior. The density of interconnections directly impacts implementation quality, as densely connected logic regions increase power consumption in physical implementation (Chandrakasan & Brodersen, 2002). Meanwhile, the depth of propagation paths serves as a critical determinant of timing performance.

**Storage Phase**. At clock edges, registers capture and maintain the results of computational processes, enabling sequential behavior and stateful operations. The bit-width of registers determines the precision of data representation, directly influenced by the accuracy needs of functional operations. It also acts as a practical indicator of operational complexity in circuit design, significantly impacting circuit performance optimization (Lee et al., 2006).

This dual-phase perspective highlights that topology is not just about combinational logic connections; it is also an intentional representation of behavioral function. Building on this idea, we propose **TopoRTL, a novel framework that explicitly captures variations in topology while maintaining the semantics of behavior**. Specifically, we design three topology-aware positional encodings that reflect the essential characteristics of storage and computation dimensions. And we utilize attention mechanisms to enable the model to recognize topological variations among different circuits. In addition, we introduce a topology-guided cross-modal alignment strategy that ensures semantic consistency between graph and textual modalities while adhering to topological constraints. This approach effectively models the intrinsic relationship between behavioral and dataflow structure.

To assess the efficacy of our proposed method, we carried out comprehensive experiments focused on PPA prediction and circuit retrieval tasks. These downstream applications are pivotal for effective

circuit optimization and generation. Our findings reveal that TopoRTL, characterized by its efficient and lightweight architecture, consistently outperforms or, at the very least, matches the performance of several advanced methodologies, including numerous large-scale language models. In addition, a detailed analysis through circuit representation visualization and further analysis robustly reinforces our central premise: *topology matters in RTL representation learning*. This research offers fresh perspectives that significantly contribute to the advancement of circuit representation learning.

## 2 RELATED WORKS AND PRELIMINARIES

In this section, we provide a systematic review of RTL representation learning approaches and present our data preprocessing pipeline. In Section 2.1, we analyze previous methods, categorizing them into behavioral methods and topological methods, while also discussing their limitations stemming from the nature of RTL. In Section 2.2, we outline our data preprocessing pipeline, which comprises two main components: register cone generation (Section 2.2.1) and multimodal data generation (Section 2.2.2).

### 2.1 RELATED WORKS

**Register Transfer Level in EDA**. Register Transfer Level (RTL) is a crucial abstraction in digital circuit design, where behavioral intent and structural topology coexist. This unique abstract level makes RTL an excellent target for circuit representation learning, which supports downstream EDA applications by reducing design time and enhancing performance.

**Behavioral Modeling for RTL**. Most approaches treat RTL as software code, focusing on learning syntax and semantics through text representations, particularly with LLMs. For instance, CodeV (Zhao et al., 2025) uses GPT-3.5 to produce natural language descriptions from Verilog code, followed by fine-tuning LLMs to enhance Verilog generation. DeepRTL (Liu et al., 2025a) presents a unified model for understanding and generating Verilog by fine-tuning CodeT5+ on a dataset linking Verilog to detailed language descriptions. DeepRTL2 (Liu et al., 2025b) extends this by combining generation with embedding-based tasks in RTL.

**Topology Modeling for RTL**. Traditional methods (Xu et al., 2022; Fang et al., 2023) for topology modeling primarily use feature engineering to transform Verilog code into graph structures, relying on hand-crafted features that may lack semantic depth and generalizability. Recently, SNS v2 (Xu et al., 2023) categorizes circuits into register cones and employs functionally equivalent contrastive learning for pretraining, using this representation for downstream tasks. However, this approach sacrifices topological awareness in the process. For instance, it cannot differentiate between Circuit B and Circuit C as shown in Figure 1.

**Multi-modal Modeling for RTL**. CircuitFusion Fang et al. (2025) pioneers multimodal representation learning for RTL by integrating code, summaries, and graphs. To capture topological information, it employs a cross-stage alignment strategy that utilizes post-synthesis netlists during pretraining, leveraging the physical implementation details to guide the behavior-aware contrastive learning process.

Overall, these approaches face significant limitations. Text-centric models often overlook the intrinsic structured nature of RTL, while traditional topological methods lack semantic generalization. Furthermore, recent multimodal attempts like CircuitFusion rely on costly logic synthesis to implicitly infer topology, limiting their efficiency and applicability in the design "left-shift" paradigm. In contrast, we propose TopoRTL, an **RTL-native** framework. Instead of depending on synthesis outcomes, TopoRTL explicitly captures topological variations directly from RTL via hardware-specific inductive biases while preserving behavioral semantics.

### 2.2 PRELIMINARIES: DATA PREPROCESSING

#### 2.2.1 REGISTER CONE GENERATION

In line with the core concept of sub-design partitioning, we extract register cones through a register-driven backward traversal. This process is outlined in Algorithm 1 and occurs in three phases. **Phase 1**. Given an RTL design $V$ with a total of registers $\{R_i\}_{i=1}^N$, we build signal dependency

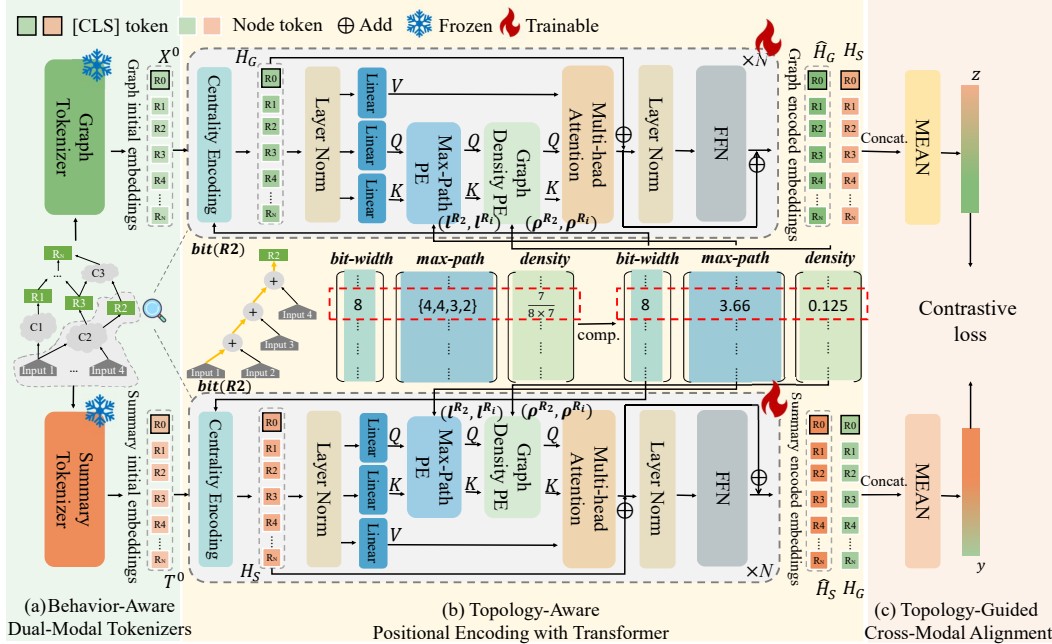

Figure 2: Overview of TopoRTL.

dictionaries that include signal declarations and combinational dependency information. **Phase 2**. We traverse the combinational logic from each register $R_i$ to its inputs/connected registers. **Phase 3**. Using the identified signals, we generate syntactically correct subcircuits $V^{R_i}$, which are verified using Yosys (Wolf et al., 2013), an open-sourced logic synthesis tool. This implementation ensures complete and scalable decomposition for RTL designs. Details are provided in Appendix D.1.

### 2.2.2 MULTIMODAL DATA GENERATION

Drawing on multimodal learning advances (Li et al., 2022a; 2021; Liu et al., 2024a; Zhao et al., 2023), we construct two modalities to explicitly modeling structural topology and behavior function: (1) **Graph modality**: we transform each subcircuit $V^{R_i}$ into a control-data flow graph (CDFG) $G^{R_i}$, where the nodes represent combinational logic and registers, while the edges encode signal connectivity. This approach is similar to the method described by Fang et al. (2025), explicitly modeling the topological structure. (2) **Summary modality**: we prompt GPT-OSS-120B (OpenAI, 2025) to generate behavioral descriptions $S^{R_i}$ capturing high-level functional intent for each subcircuit $V^{R_i}$. This dual-representation framework enhances circuit behavior and topology learning.

## 3 METHODOLOGY

We introduce TopoRTL, a framework that integrates behavior functions with topology structure information. As illustrated in Figure 2, TopoRTL has three key components: (1) Behavior-Aware Dual-Modal Tokenizers for extracting semantics from topology graphs and functional descriptions; (2) Topology-Aware Positional Encoding that incorporates bit-width centrality, signal path depth, and interconnection density into Transformer attention; and (3) Topology-Guided Cross-Modal Alignment that merges modalities while maintaining topological constraints. The representations generated by TopoRTL can be applied to tasks such as PPA prediction and circuit retrieval.

### 3.1 BEHAVIOR-AWARE DUAL-MODAL TOKENIZERS

To capture the behavior information of circuits, we utilize behavior-aware dual-modal tokenizers that are trained through a behavior equivalence contrastive learning task and a mask modeling task.

**Graph-Based Tokenizer.** To capture topology-aware circuit semantics, we employ a pretrained graph tokenizer that maps sub-circuits to compact latent representations. For a design decomposed

into $N$ sub-circuits $\{G^{R_i}\}_{i=1}^N$, the tokenizer outputs a representation $x^{R_i} \in \mathbb{R}^{1 \times d}$ for each sub-circuit $G^{R_i}$. These representations are combined with a global design-level [CLS] token $x^{R_0}$ to form the input sequence for downstream tasks:

$$X^0 = (x^{R_0\,T}, x^{R_1\,T}, \dots, x^{R_N\,T})^T \in \mathbb{R}^{(1+N) \times d}. \tag{1}$$

This sequence preserves hierarchical design semantics while enabling efficient processing by transformer-based models. For more details, please refer to Appendix D.2.1.

**Summary-Based Tokenizer.** To capture behavioral semantics from circuit descriptions, we employ a pretrained summary tokenizer based on BERT that encodes textual summaries into semantic embeddings. For a design with $N$ sub-circuits and their textual summaries $\{S^{R_i}\}_{i=1}^N$, the tokenizer outputs a global [CLS] token embedding $t^{R_i} \in \mathbb{R}^{1 \times d}$. These embeddings are combined with a learnable global design-level [CLS] token $t^{R_0}$ to form the input sequence:

$$T^0 = (t^{R_0\,T}, t^{R_1\,T}, \dots, t^{R_N\,T})^T \in \mathbb{R}^{(1+N) \times d}. \tag{2}$$

This sequence enables transformer models to jointly reason over circuit functionality. For more details, please refer to Appendix D.2.2.

### 3.2 Topology-Aware Positional Encoding with Transformer

#### 3.2.1 Bit-Width Centrality Encoding

During the storage phase, registers preserve computational results where bit-width directly determines the precision range of data representation. In practice, complex operations (e.g., 32-bit arithmetic units) inherently require wider bit-widths to maintain accuracy, while simpler control signals (e.g., 1-bit flags) operate effectively with minimal precision (Lee et al., 2006). To enable the model to distinguish such functional hierarchies from circuit topology, we propose *Bit-Width Centrality Encoding*.

**Bit-width Encoding**. For each register $R_i$, we extract $bit(R_i)$ from Verilog declarations (e.g., `reg [31:0] data;`) to encode precision constraints as topology features. We first process the initial node features $X^0$ and $S^0$ from dual modalities through a multi-layer perception (MLP):

$$X = \text{MLP}(X^0) \in \mathbb{R}^{(1+N) \times d}, \quad S = \text{MLP}(S^0) \in \mathbb{R}^{(1+N) \times d}, \tag{3}$$

where $X, S \in \mathbb{R}^{(1+N) \times d}$ and $N$ denotes the total number of registers and $d$ is the feature dimension.

Subsequently, we assign two learnable embedding vectors $a_G^{bit}$ and $a_S^{bit}$ for each possible bit-width value. These embedding vectors are accessed through a lookup table mechanism based on each register's actual bit-width:

$$h_G^{R_i} = x^{R_i} + a_G^{bit(R_i)}, \quad h_S^{R_i} = s^{R_i} + a_S^{bit(R_i)} \quad 1 \le i \le N, \tag{4}$$

where $x^{R_i}$ and $s^{R_i}$ are the features after MLP processing, and $a_G^{bit(R_i)}$ and $a_S^{bit(R_i)}$ are the learnable embedding vectors corresponding to the bit-width of register $R_i$. This positional encoding method helps the model associate bit-width values with functional complexity during topological learning.

#### 3.2.2 Max-Path and Density Discrepancy Encoding

During the computation phase, signals traverse through combinational logic networks, where high interconnection density raises power consumption due to increased parasitic capacitance (Chandrakasan & Brodersen, 2002). The propagation path depth also influences timing performance through the critical path length. To help the model differentiate these structural factors from circuit topology, we introduce *Max-Path and Density Discrepancy Encoding*.

**Max-Path Encoding**. For each register cone $G^{R_i}$, where $1 \le i \le N$, we extract the maximum path length set:

$$L^{R_i} = \{dist(R_i, R_j) \mid \text{exist path } R_j \to R_i \text{ in } G^{R_i}\} \quad 1 \le i \le N, \tag{5}$$

where $dist(R_j, R_i)$ represents the number of pseudo logic gates between registers $R_j$ and $R_i$. Rather than relying solely on the absolute maximum path length, which can be sensitive to outliers, we select the Top-K longest paths and compute their mean for robust representation:

$$l^{R_i} = \text{MEAN}(\text{Top-K}(L^{R_i})) \quad 1 \le i \le N. \tag{6}$$

This approach captures the typical critical path behavior while mitigating the impact of anomalous paths. We then construct a relative matrix $\Delta L \in \mathbb{R}^{N \times N}$, where

$$\Delta L_{ij} = |l^{R_i} - l^{R_j}| \quad 1 \le i, j \le N, \tag{7}$$

representing the discrepancy in critical path characteristics between register pairs.

**Graph Density Encoding**. For each $G^{R_i}$, we compute graph density as:

$$\rho^{R_i} = \frac{E^{R_i}}{N^{R_i}(N^{R_i} - 1)} \quad 1 \le i, j \le N, \tag{8}$$

where $E^{R_i}$ and $N^{R_i}$ denote the number of edges and nodes in the register cone, respectively. This metric quantifies how interconnected the logic surrounding register $R_i$ is, with higher values indicating more complex, tightly coupled functionality. We then compute a relative density discrepancy matrix $\Delta \rho \in \mathbb{R}^{N \times N}$, where

$$\Delta \rho_{ij} = |\rho^{R_i} - \rho^{R_j}| \quad 1 \le i, j \le N. \tag{9}$$

### 3.2.3 TRANSFORMER WITH TOPOLOGY-AWARE ATTENTION

The Transformer architecture consists of a composition of Transformer layers, each containing two key components: a self-attention module and a position-wise feed-forward network (FFN). To illustrate our approach, we specifically describe the process using the graph modality $H_G$. Here, $H_G$ serves as the input to the self-attention module with hidden dimension $d$, where each position represents the $i$-th register in the RTL circuit. This input is projected into three matrices through learnable weight parameters $W_G^Q \in \mathbb{R}^{d \times d_K}$, $W_G^K \in \mathbb{R}^{d \times d_K}$, and $W_G^V \in \mathbb{R}^{d \times d_V}$ to obtain the corresponding representations $Q_G, K_G, V_G$:

$$Q_G = H_G W_G^Q, \quad K_G = H_G W_G^K, \quad V_G = H_G W_G^V, \tag{10}$$

$$A_G = \frac{Q_G K_G^T}{\sqrt{d_K}}, \quad Attn(H_G) = \text{softmax}(A_G)V_G, \tag{11}$$

where $A_G$ captures the similarity between queries and keys. For clarity, we consider the single-head self-attention mechanism, assuming that $d_K = d_V = d$. This analysis is presented in the context of graph modality, where the summary modality is the same.

The vanilla Transformer architecture is powerful for sequential data but fails to account for the unique topological properties of RTL circuits. Unlike linear natural language sequences, RTL circuits have complex hierarchical structures where signal paths and connection densities are crucial for functionality. To overcome this limitation, we integrate our previously proposed *Max-Path and Density Discrepancy Encodings* into the attention mechanism:

$$A_{Gij} = \frac{(h_G^{R_i} W_G^Q)(h_G^{R_j} W_G^K)^T}{\sqrt{d}} + \alpha_G \cdot f_G(\Delta L_{ij}) + \beta_G \cdot g_G(\Delta \rho_{ij}), \tag{12}$$

where $f_G(\cdot), g_G(\cdot) : \mathbb{R} \to \mathbb{R}^{1 \times n_{head}}$ are learnable mapping functions implemented as MLPs, and $\alpha_G, \beta_G$ are learnable scaling parameters, and $1 \le i, j \le N$. This formulation enables the attention mechanism to dynamically adjust its focus based on both the timing characteristics and structural complexity of register relationships.

For the virtual node $R_0$ representing the entire circuit, we manage its connections uniquely by resetting all spatial encodings to distinct learnable scalars. The final circuit representation is produced by processing the inputs through modified Transformer layers:

$$\hat{H}_G = \text{Graph-Transformer}(H_G), \quad \hat{H}_S = \text{Summary-Transformer}(H_S). \tag{13}$$

### 3.3 TOPOLOGY-GUIDED CROSS-MODAL ALIGNMENT

Achieving effective alignment across various modalities is essential for a thorough understanding of circuit representation learning. To enhance the model's ability to comprehend circuit topology,

Table 1: PPA prediction results, and model specifications. The best, second-best, and third-best results in each column are highlighted with **bold**, underlined, and *italic* fonts, respectively.

| Method | Type | Size | Circuit Data | Dim | Area | | | | Power | | | |
|---|---|---|---|---|---|---|---|---|---|---|---|---|
| | | | | | PCC↑ | $R^2$↑ | MAPE↓ | RRSE↓ | PCC↑ | $R^2$↑ | MAPE↓ | RRSE↓ |
| GCN-MLP | Graph | 1.20M | 7k | 768 | 0.507 | -0.934 | 20.878 | 0.873 | 0.378 | -7.217 | 55.454 | 0.929 |
| GCN-GNN | Graph | 1.20M | 7k | 768 | 0.340 | -80.062 | 17.569 | 0.969 | 0.314 | -82.125 | 50.316 | 0.972 |
| Qwen3-E-0.6B | Text | 0.6B | - | 1024 | 0.694 | 0.422 | 13.735 | 0.858 | 0.743 | 0.515 | 37.917 | 0.796 |
| Qwen3-E-4B | Text | 4B | - | 2560 | 0.760 | 0.560 | 11.541 | 0.753 | 0.716 | 0.382 | 38.341 | 0.939 |
| Qwen3-E-8B | Text | 8B | - | 4096 | 0.720 | 0.451 | 12.079 | 0.876 | 0.766 | 0.556 | 37.826 | 0.821 |
| CodeV-CL | Text | 7B | 165k | 4096 | 0.795 | 0.596 | 11.574 | 0.661 | *0.812* | *0.633* | 39.448 | 0.623 |
| CodeV-DS | Text | 6.7B | 165k | 4096 | *0.814* | *0.637* | 10.778 | 0.626 | 0.827 | 0.673 | 36.544 | *0.624* |
| CodeV-QC | Text | 7B | 165k | 3584 | 0.818 | 0.662 | *10.830* | *0.648* | 0.805 | 0.622 | *37.314* | 0.678 |
| CircuitFusion | Multi | 150.59M | 7k | 768 | 0.647 | 0.378 | 13.242 | 1.085 | 0.657 | 0.393 | 43.073 | 0.993 |
| TopoRTL | Multi | 29.13M | 7k | 768 | **0.863** | **0.683** | **7.952** | **0.574** | **0.884** | **0.712** | **25.033** | **0.585** |

| | Slack | | | | TNS | | | | WNS | | | |
|---|---|---|---|---|---|---|---|---|---|---|---|---|
| | PCC↑ | $R^2$↑ | MAPE↓ | RRSE↓ | PCC↑ | $R^2$↑ | MAPE↓ | RRSE↓ | PCC↑ | $R^2$↑ | MAPE↓ | RRSE↓ |
| GCN-MLP | 0.708 | 0.500 | 42.414 | 0.945 | 0.733 | 0.510 | 41.095 | 0.793 | 0.679 | 0.425 | 47.647 | 0.903 |
| GCN-GNN | 0.532 | 0.243 | 52.620 | 1.347 | 0.889 | 0.770 | 33.430 | 0.498 | 0.634 | 0.336 | 47.506 | 0.952 |
| Qwen3-E-0.6B | 0.876 | 0.724 | 35.587 | 0.554 | 0.885 | 0.753 | *30.944* | 0.555 | 0.860 | *0.667* | *40.477* | 0.728 |
| Qwen3-E-4B | 0.881 | 0.753 | 35.162 | 0.570 | 0.884 | 0.777 | 39.324 | 0.520 | 0.839 | 0.686 | 52.680 | 0.718 |
| Qwen3-E-8B | *0.888* | 0.784 | 34.241 | 0.563 | 0.899 | 0.781 | 33.802 | 0.534 | *0.849* | 0.659 | 43.880 | 0.674 |
| CodeV-CL | **0.909** | **0.822** | **30.472** | 0.465 | *0.922* | *0.846* | **28.108** | *0.428* | 0.806 | 0.643 | 41.267 | *0.716* |
| CodeV-DS | 0.881 | 0.758 | 32.712 | 0.579 | **0.928** | 0.848 | 31.857 | **0.383** | 0.780 | 0.600 | 41.750 | 0.735 |
| CodeV-QC | 0.868 | 0.754 | 34.618 | 0.575 | 0.927 | **0.856** | 29.920 | 0.402 | 0.762 | 0.464 | 47.401 | 1.400 |
| CircuitFusion | 0.893 | *0.788* | 30.944 | *0.494* | 0.885 | 0.727 | 34.454 | 0.544 | 0.817 | 0.572 | **38.227** | 0.808 |
| TopoRTL | **0.909** | 0.821 | *31.249* | **0.443** | 0.872 | 0.743 | 32.016 | 0.521 | **0.862** | **0.723** | 40.130 | **0.580** |

we introduce a topology-guided cross-modal alignment mechanism. This innovative approach capitalizes on our previously encoded structural information, ensuring that meaningful correspondences are established between modalities while honoring the inherent topology of the circuits.

Let $Y = (H_G, \hat{H}_S) \in \mathbb{R}^{(1+N) \times 2d}$ and $Z = (\hat{H}_G, H_S) \in \mathbb{R}^{(1+N) \times 2d}$ represent two complementary fusion patterns between the graph modality $(H_G, \hat{H}_G)$ and summary modality $(H_S, \hat{H}_S)$, where $N$ is the number of registers and $d$ is the feature dimension. We compute their global representations by taking the mean across nodes:

$$y = \text{MEAN}(Y) \in \mathbb{R}^{1 \times 2d}, \quad z = \text{MEAN}(Z) \in \mathbb{R}^{1 \times 2d}. \tag{14}$$

Our topology-guided approach uses structural constraints to align $y$ and $z$ while maintaining circuit topological properties. We employ a quadruplet loss that pulls positive pairs closer and ensures topological consistency by requiring the difference between $y$ and $z$ to be smaller than that of embeddings from topologically dissimilar paths. Negative samples are randomly selected as graph modality fused embedding $y_{neg}$ and summary modality fused embedding $z_{neg}$ from the batch. The contrastive learning loss is:

$$\mathcal{L}_{fuse} = [\| y - z \|_2^2 - \| y - z_{neg} \|_2^2 + \beta]_+ + [\| z - y \|_2^2 - \| z - y_{neg} \|_2^2 + \beta]_+, \tag{15}$$

where $\beta$ is a hyperparameter that controls the margin of the distance between pairs of positive and negative samples, and $[\cdot]_+$ denotes $\max(0, \cdot)$. This loss serves as the pretraining loss.

## 4 EXPERIMENTS

In this section, we conduct experiments to address the following research questions:

- **RQ1:** How does TopoRTL excel in topology-dependent tasks? Does it effectively capture essential topological dependencies for precise predictions?

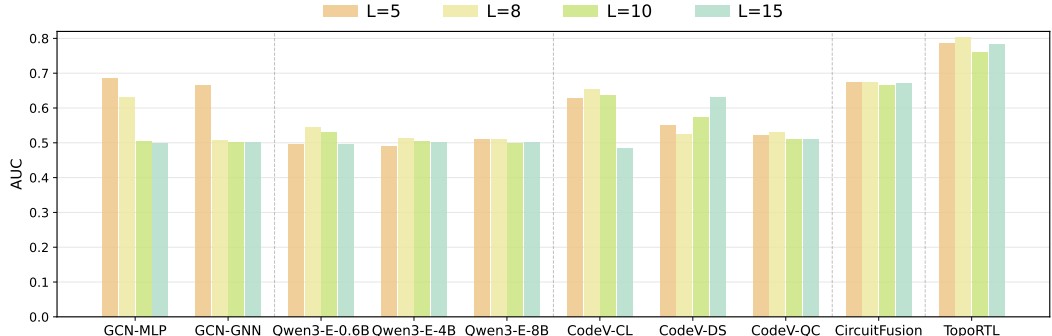

Figure 3: Circuit Retrieve Performance.

- **RQ2:** How well does TopoRTL integrate topological structure in behavior-sensitive tasks? Can it overcome the topological neglect seen in existing methods?
- **RQ3:** Do TopoRTL embeddings maintain both local structural details and global topological relationships in hidden spaces?
- **RQ4:** What unique contributions do its encodings make to representation quality?

## 4.1 EXPERIMENTAL SETUP

We begin by briefly outlining the dataset, baseline methods, and the evaluation tasks and metrics. For more detailed descriptions of the experimental settings, please refer to Appendix B and C.

**Evaluation Tasks and Metrics**. To evaluate the capability of RTL representation learning, we selected two downstream tasks: **Performance, Power, Area (PPA) Prediction** and **Natural Language Code Search**. The first is a regression task, using evaluation metrics such as **PCC**, $R^2$, **MAPE**, and **RRSE**. The second task is framed as a retrieval classification (Lu et al., 2021), with **AUC** as the evaluation metric. For further details, please refer to Appendix C.

**Circuit Dataset**. We construct a dataset with **115** RTL designs collected from OpenCores (Albrecht, 2005), VexRiscv (Papon & Spinal, 2024), ITC'99 (Corno et al., 2002), and DeepCircuitX (Li et al., 2025). The circuit dataset has a wide range of circuit sizes, with different scales and functions. After extracting register cones, the dataset consists of **7,576** sub-circuits. For more information on data collection, processing, and statistics, please refer to Appendix B.

**Baseline Models and Implementation Details**. We compare TopoRTL with baselines in three categories. **(i) Graph modality models**: Graph Convolutional Networks (GCN) with two types of finetune methods, e.g., GCN-MLP and GCN-GNN. **(2) Text modality models**: Open-source models Qwen3-Embedding (abbreviated as Qwen3-E) (Zhang et al., 2025) and Verilog-specialized CodeV (Zhao et al., 2025). CodeV includes three variants: CodeV-CL-7B, CodeV-DS-6.7B, CodeV-QC-7B. **(2) Multimodal models**: CircuitFusion (Fang et al., 2025). For more baseline information and implementation details, please refer to Appendix C.4 and C.5.

## 4.2 PERFORMANCE ON PPA PREDICTION (RQ1)

To assess the ability to represent topology information, we performed five PPA prediction tasks covering Slack, Worst Negative Slack (WNS), Total Negative Slack (TNS), Area, and Power metrics. Further details about PPA tasks can be found in Appendix C.1, while the experimental analysis area is detailed in Appendix D.3.1. Based on Tables 1, we can draw the following observations:

- **Obs 1: TopoRTL achieves holistic RTL modeling superiority through topology-behavior integration with lightweight architecture**. Specifically, it dominates ppa metrics (↑ 5.5% Area PCC, ↑ 6.9% Power PCC, ↓ 26.2% Area MAPE, ↓31.5% Power MAPE) and sets the timing benchmark (WNS PCC=0.862, RRSE=0.580), outperforming all baselines in critical-path analysis while matching Slack accuracy. Crucially, these improvements come with fewer parameters and training data, showcasing TopoRTL's effectiveness in capturing global topological dependencies that text-based models struggle with.

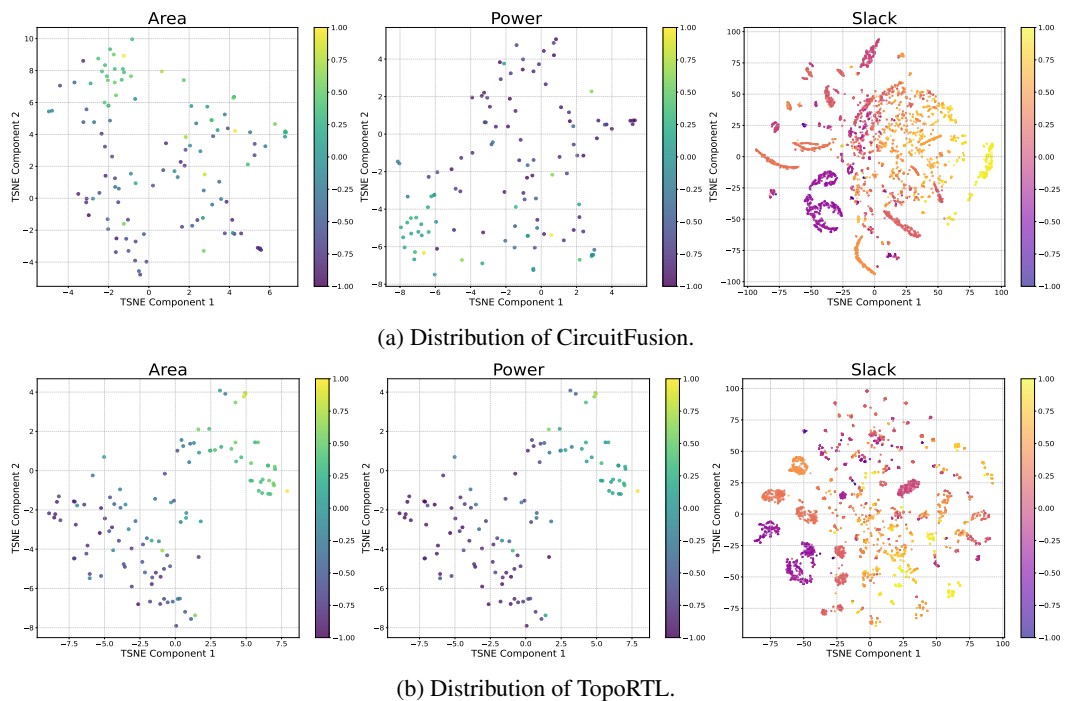

(a) Distribution of CircuitFusion.

(b) Distribution of TopoRTL.

Figure 4: The distribution of hidden representations across different models and tasks.

## 4.3 PERFORMANCE ON CIRCUIT SEARCH (RQ2)

To evaluate behavioral representation capabilities, we conduct a natural language code search task critical for hardware design reuse and verification. Following Lu et al. (2021), we evaluate with $L$ negative designs ($L \in \{5, 8, 10, 15\}$) per query, measuring performance via AUC. Further details regarding this task can be found in Appendix C.2, while the analysis of detailed experiment results is presented in Appendix D.3.2. Based on Figure 3, we derive the key insight below:

- **Obs 2: TopoRTL demonstrates superior performance and robustness across retrieval scenarios**. Our model maintains a stable performance near 0.8 AUC for all $L$ values (5-15 negative samples), outperforming all baselines. This consistency stems from TopoRTL's joint modeling of topology and behavior, emphasizing the importance of topology in RTL representation learning. The topology-guided alignment mechanism filters out irrelevant samples, ensuring reliable behavioral matching even in noisy conditions, thus enhancing cross-modal retrieval accuracy and supporting scalable design reuse across various hardware applications.

## 4.4 HIDDEN REPRESENTATIONS ANALYSIS (RQ3)

As demonstrated in the previous sections, TopoRTL effectively learns both topological and behavioral circuit characteristics. To further validate this, we visualize the learned representations using t-SNE (Maaten & Hinton, 2008). Embeddings are projected into 2D space, colored by normalized Area, Power, and Slack metrics. The analysis of detailed experiment results is presented in Appendix D.3.3. According to Figure 4, we can find that:

- **Obs 3: TopoRTL produces well-structured embeddings that clearly distinguish between topologically diverse regions**. The representations illustrate both continuous topological trends and distinct design regimes. For *Area* and *Power*, smooth gradients show how circuit scale systematically corresponds to embedding position. In contrast, for *Slack*, non-overlapping clusters align with critical-path topologies. However, CircuitFusion produces fragmented distributions, characterized by abrupt value jumps and severe cluster entanglement, which fail to maintain either continuity or clear separation between design regimes. Additionally, isolated outliers within TopoRTL's Slack clusters suggest unresolved mismatches between RTL and gate-level implemen-

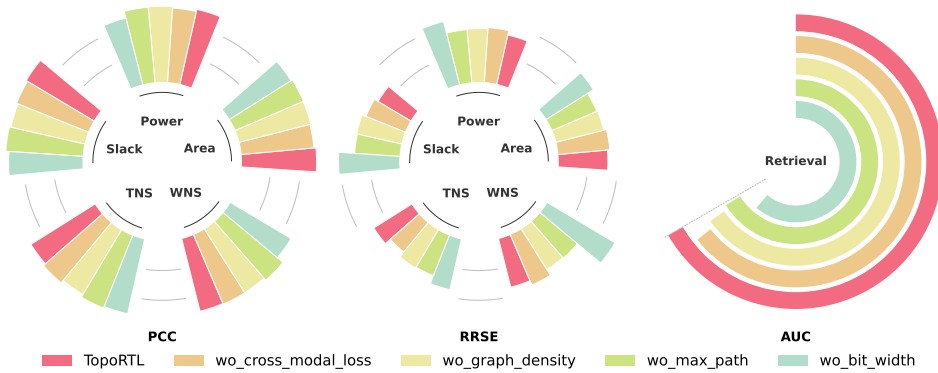

Figure 5: Ablation Study.

tations, resulting from abstraction loss and hierarchical misalignment. This points to opportunities for further refining the topology features.

## 4.5 ABLATION AND FURTHER ANALYSIS (RQ4)

**Abalation Study**. To validate the contribution of each TopoRTL component, we conduct comprehensive ablation experiments by systematically removing key modules. More details and analysis are provided in Appendix D.3.4. As shown in Figure 5, these experiments reveal:

- **Obs 4: Positional encodings improve performance across tasks**. Bit-width encoding effectively captures topology and complexity, while max-path and density encodings show inconsistent results, highlighting the need for complementary topological signals in circuit representation.
- **Obs 5: Topology-guided alignment favors topology fidelity**. This approach prioritizes topology-semantic consistency, which may slightly reduce timing accuracy but significantly boosts other topological and behavioral tasks, underlining its importance for design optimization.

**We recommend readers check Appendix D.3 for detailed experiments and analysis.**

## 5 LIMITATION AND FUTURE DISCUSSION

While TopoRTL demonstrates significant improvements in RTL representation learning, several limitations warrant attention. First, scaling to larger and more diverse RTL datasets would enhance the model's generalization across circuit architectures. Second, our current decomposition approach assumes synchronous sequential circuits and disrupts clock domain relationships during register cone extraction; future work should extend to handle asynchronous circuits through clock-aware decomposition strategies. Additionally, developing more sophisticated topology-aware positional encodings could better capture complex signal propagation patterns. Addressing these limitations would further strengthen the framework's applicability to practical chip design scenarios.

## 6 CONCLUSION

In this work, we analyze RTL circuits that fundamentally operate as structured dataflow graphs where behavioral semantics and topological structure are inseparable. Inspired by this, we propose TopoRTL, a novel framework that explicitly encodes topological relationships while preserving behavioral functionality. Specifically, we develop dual modalities that are initialized using behavior-aware tokenizers and create three topology-aware positional encodings grounded in signal propagation. Additionally, we introduce a topology-guided cross-modal alignment strategy, enhancing the integration and interaction between the modalities. Extensive experiments across ppa and retrieval tasks definitively demonstrate TopoRTL's superiority in jointly capturing topological and behavioral characteristics, proving that *topology matters in RTL representation learning*.

ETHICS STATEMENT

This work enhances representation learning for RTL circuits to improve automated chip design. Our research aims for more efficient hardware development, potentially leading to energy savings and advanced computational capabilities. While focusing on circuit representation, we acknowledge the broad societal implications of chip design automation. We adhere to the ICLR Code of Ethics, ensuring rigorous experimentation and accurate reporting of results. Our datasets consist of standard benchmark circuits, with no personal information or human subjects involved. We urge the chip design community to consider environmental impacts, maintain human oversight, and promote transparency in AI-assisted design systems, committing to responsible research for societal well-being.

REPRODICIBILITY STATEMENT

To ensure reproducibility, we provide detailed descriptions of our methodology and experimental setup. All circuit datasets are sourced from open-source benchmarks, which provide complete documentation of data processing procedures and statistical characteristics. These details can be found in Appendix B and Appendix C. Additionally, the source code is publicly available at the following URL: `https://github.com/BUPT-GAMMA/TopoRTL.git`.

ACKNOWLEDGMENTS

This work is supported by the National Natural Science Foundation of China (No. 62550138, 62192784, 62572064, 62472329, 62404021), the Beijing Natural Science Foundation (No. 4244107, QY24204), the State Key Lab of Processors, Institute of Computing Technology, CAS (No. CLQ202504), the Fundamental Research Funds for the BUPT (No. 2025AI4S20), and the Research Initiation Project for Introduced Talents of BUPT (No. 510224062).

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

# Appendix Table of Contents

Table 2: Circuit Benchmarks Statistics

| Source Benchmarks | #Circuit | Circuit Size (Min, Avg, Max) | | |
|---|---|---|---|---|
| | | #Gate | #Token (Code) | #Register |
| ITC'99 | 18 | (135, 5K, 22K) | (2K, 284K, 262K) | (5, 45.0, 252) |
| OpenCores | 12 | (360, 5K, 28K) | (1K, 182K, 1M) | (7, 59.8, 371) |
| VexRiscv | 13 | (7K, 14K, 63K) | (112K, 232K, 1M) | (67, 141.2, 434) |
| DeepCircuitX | 72 | (64, 4K, 66K) | (187, 53K, 1M) | (1, 58.5, 1326) |
| Total | 115 | 711K | 14M | 7576 |

# A    THE USE OF LARGE LANGUAGE MODELS

In preparing this manuscript, we utilized Large Language Models (LLMs) solely as a general-purpose writing assistance tool for minor language refinement and grammatical correction. Specifically, we utilized LLMs to identify basic syntax errors, enhance sentence clarity, and ensure proper academic phrasing in non-technical sections of the text. We carefully reviewed and verified all content produced with LLM assistance to ensure accuracy and maintain scientific integrity. We are responsible for all content in this manuscript, following ICLR's policies on LLM usage.

# B    DATASET DETAILS

## B.1    SOURCE BENCHMARKS

In this section, we provide an overview of the various hardware description languages (HDLs) circuit datasets used in this work.

### B.1.1    ITC'99

The ITC'99 (Corno et al., 2002) benchmark circuits represent a standardized set of circuits with characteristics typical of synthesized designs. As one of the established unimodal benchmark datasets alongside ISCAS'89 and EPFL, it continues to serve as an important resource for circuit verification and testing methodologies.

### B.1.2    OPENCORES

OpenCores (Albrecht, 2005) is a prominent online community established in 1999 for the development and sharing of gateware Intellectual Property (IP) cores. It serves as a collaborative platform where digital designers can showcase, promote, and discuss their work through forums and news channels. The OpenCores repository hosts diverse RTL designs, including DSP cores, crypto cores, memory cores, and various system-level implementations. As one of the largest open-source hardware communities, it provides a version control system for source management and supports a vibrant user community dedicated to free and open-source hardware collaboration.

### B.1.3    VEXRISCV

VexRiscv (Papon & Spinal, 2024) is an FPGA-friendly 32-bit RISC-V CPU implementation. VexRiscv supports M, C, and A RISC-V instruction set extensions with numerous optimizations, including multi-stage pipelines and data caching capabilities. Implemented in SpinalHDL, VexRiscv utilizes complementary plugins to enhance functionality while maintaining a streamlined core architecture, making it particularly suitable for FPGA-based system-on-chip designs.

### B.1.4    DEEPCIRCUITX

DeepCircuitX (Li et al., 2025) represents a holistic, repository-level dataset specifically curated to address limitations in existing RTL datasets. It provides comprehensive data and annotations across multiple abstraction levels, like chip, IP, module, and RISC-V. The dataset features multi-level source RTL code spanning repository, file, module, and block levels, with corresponding annotations

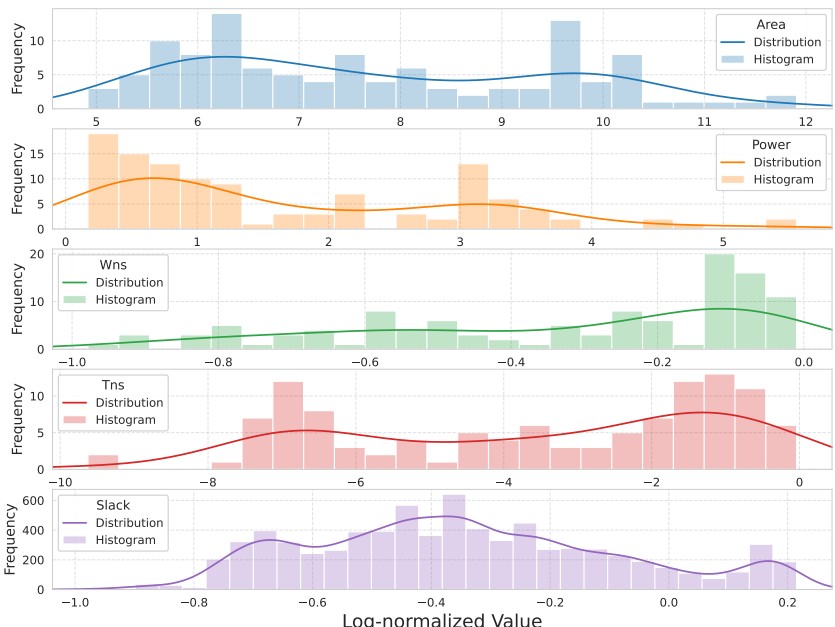

Figure 6: Label Distribution Statistics

generated by GPT-4o. It establishes specialized benchmarks for RTL understanding, generation, and completion tasks, with detailed data distributions across different RTL categories as documented in its comprehensive dataset summaries.

### B.2 DATASET PROCESS

This section details our data processing methodology and label generation approach for different downstream tasks. We first selected 115 syntactically correct sequential circuits from the aforementioned four open-source benchmarks that can be directly synthesized. We then generated task-specific labels for PPA prediction and circuit retrieval tasks.

**PPA Label Generation**. To address the heterogeneity of HDLs across different sources, including VHDL, Verilog, and SpinalHDL, we employed Yosys to standardize all designs into a unified Verilog representation. Subsequently, we utilized Synopsys Design Compiler, an industry-standard logic synthesis tool, to automatically synthesize each RTL circuit into gate-level netlists. These netlists represent the actual circuit implementations composed of logic gates (e.g., ADD, INV, AND, etc.) and registers (DFF) from a specific technology library. The synthesis process employed the open-source NanGate 45nm standard cell library, with the `compile_ultra` command to ensure high-quality PPA metrics on the Pareto frontier, as verified by Fang et al. (2023). Finally, Synopsys PrimeTime was utilized to analyze the gate-level netlists, extracting detailed PPA labels, which include timing metrics such as Slack, WNS, and TNS, as well as measurements for Area and Power. The statistics related to the RTL designs post-synthesis are presented in Table 2. Additionally, the distribution of labels can be found in Figure 6.

**Query Generation**. For natural language code retrieval experiments, we developed a two-stage query generation pipeline using large language models (LLMs) followed by embedding encoding. First, we prompted the LLM to generate detailed descriptions for each module within a circuit, covering its name, inputs, outputs, functionality, and sub-module instantiations. Second, we concatenated all module descriptions from the same circuit and prompted the LLM to produce a high-level functional summary that mimics human retrieval behavior. This two-stage approach offers two significant advantages: (1) it effectively mitigates the context window limitations of LLMs through modular processing, and (2) the resulting high-level circuit summaries present a more challenging test for circuit representation models, better evaluating their ability to capture semantic functionality rather than merely syntactic patterns. Here we use GPT-OSS-120B to obtain descriptions

and Qwen3-Embedding-8B to embed them. For the prompts we use to generate module-level and design-level descriptions, please refer to Appendix G.

## C Evaluation Details

This section first introduces the two downstream tasks for evaluating pre-trained models, PPA prediction and natural language code retrieval, along with our unified evaluation framework. We then detail the selected baselines and their parameter configurations.

### C.1 PPA Prediction Task

The Performance, Power, and Area (PPA) prediction task represents a critical design quality evaluation at the RTL stage, enabling early assessment of circuit implementation characteristics without full synthesis. We evaluate five key prediction tasks:

- **Register slack prediction**: forecasting timing margins for individual registers, which identifies potential timing violation points in the circuit.
- **WNS prediction**: estimating the Worst Negative Slack, representing the most severe timing violation across the entire design.
- **TNS prediction**: predicting the Total Negative Slack, which aggregates all timing violations to indicate overall timing quality.
- **Power prediction**: assessing the circuit's power consumption for energy efficiency evaluation.
- **Area prediction**: determining the silicon footprint required for implementation, crucial for physical feasibility and cost considerations.

Notably, register slack prediction operates at the sub-circuit level, while the remaining four metrics are evaluated at the complete circuit level.

**Metric**. We employ four complementary metrics to comprehensively assess prediction quality:

- **PCC**: Pearson correlation coefficient, which assesses the linear correlation between predictions and ground truth. Formally, given the prediction value vector $x$ and the truth label $y$, it is calculated as follows:
$$PCC = \frac{\sum (x - m_x)(y - m_y)}{\sqrt{\sum (x - m_x)^2 \sum (y - m_y)^2}},$$
where $m_x$ is the mean of $x$ and $m_y$ is the mean of $y$. The metric varies between $-1$ and $1$.
- $R^2$: Coefficient of determination, which measures the proportion of variance explained by the model. Formally, prediction value $x$ and the truth label $y$ with $n$ samples, it is calculated as follows:
$$R^2 = 1 - \frac{\sum_{i=1}^{n}(y_i - x_i)^2}{\sum_{i=1}^{n}(y_i - \bar{y})^2},$$
where $\bar{y} = \frac{1}{n}\sum_{i=1}^{n} y_i$. The best value score is 1.0, and it can be negative (because the model can be arbitrarily worse).
- **MAPE**: Mean absolute percentage error, which Quantifies prediction error as a percentage of ground truth. Formally, prediction value $x$ and the truth label $y$ with $n$ samples, it is calculated as follows:
$$MAPE = \frac{100\%}{n} \sum_{i=1}^{n} |\frac{x_i - y_i}{y_i}|.$$
This metric is nonnegative, and the lower the better.
- **RRSE**: Root relative squared error, which is a commonly used regression metric to measure the prediction error. Formally, prediction value $x$ and the truth label $y$ with $n$ samples, it is calculated as follows:
$$RRSE = \sqrt{\frac{\sum_{i=1}^{N}(x_i - y_i)^2}{\sum_{i=1}^{N}(y_i - \bar{y})^2}},$$
where $\bar{y} = \frac{1}{n}\sum_{i=1}^{n} y_i$.

This multi-metric approach provides a balanced evaluation, capturing both correlation strength and absolute prediction accuracy.

## C.2    NATURAL LANGUAGE CODE SEARCH

Natural language code search enables hardware designers to locate relevant RTL implementations through intuitive natural language queries, significantly enhancing design productivity and code reuse. This task involves embedding both natural language queries and circuit implementations into a shared semantic space, where relevance is determined by vector similarity. For hardware design contexts, this capability is particularly valuable as it bridges the gap between high-level specifications and concrete RTL implementations, accelerating the design process and reducing manual search effort.

**Metric**. Following Lu et al. (2021), we formulate this as a retrieval classification problem. For each query, we sample $L$ negative circuit designs and measure ranking quality using **AUC (Area Under the ROC Curve)**, a robust information retrieval metric that evaluates the model's ability to distinguish relevant from irrelevant designs across all possible classification thresholds. AUC values range from 0 to 1, with higher scores indicating superior retrieval performance, where 1.0 represents perfect ranking and 0.5 indicates random performance.

## C.3    EVALUATION FRAMEWORK

To ensure fair and rigorous evaluation across diverse representation models, we implement a standardized assessment framework with strict separation of training, validation, and test phases. Our methodology proceeds as follows: First, models undergo pre-training on unlabeled RTL circuits, with hyperparameters carefully adhering to original publications to maintain implementation fidelity. After pre-training completion, we systematically extract circuit representations from multiple training epochs. For each downstream task, we then fine-tune a consistent classification/regression head architecture using these representations, with the optimal pre-training checkpoint selected exclusively based on validation set performance. Crucially, the test set remains completely isolated throughout both pre-training and fine-tuning processes, guaranteeing unbiased evaluation.

This approach offers two significant advantages: (1) it decouples representation quality from downstream task optimization, providing a cleaner assessment of learned representations; and (2) it ensures fair comparison by standardizing the fine-tuning process across all models. Crucially, the test set remains completely isolated throughout both pre-training and fine-tuning phases, guaranteeing unbiased performance evaluation.

## C.4    BASELINES

We evaluate TopoRTL against a comprehensive set of representative baselines spanning three fundamental paradigms in circuit representation learning. These baselines were strategically selected to address critical research questions:

1. Can conventional graph-based approaches effectively capture RTL topology?
2. Can text-based models overcome their inherent limitations when processing structured RTL circuits?
3. How do existing multimodal frameworks integrate topological and behavioral information?

By comparing against these diverse approaches, we establish a rigorous evaluation framework that isolates the specific contributions of TopoRTL's topology-aware architecture while addressing the fundamental question of whether explicit topological modeling provides measurable advantages over conventional representation learning methods.

**Graph Modality Models**. Graph Convolutional Network (GCN) (Kipf & Welling, 2017) has demonstrated success in general graph representation tasks. Following the methodology established by Xu et al. (2023), we implement a 3-layer GCN pre-training on functional equivalence contrastive learning tasks. Notably, Xu et al. (2023) employs a hierarchical graph structure that constructs register dataflow graphs based on inter-subgraph connections during downstream tasks. To ensure both methodological fidelity and evaluation consistency, we implement two variants: **GCN-GNN**, which preserves the original hierarchical approach with graph-based fine-tuning; **GCN-MLP**, which aligns with our unified evaluation framework by replacing hierarchical processing with a standard MLP head. This baseline specifically tests whether topology alone, without explicit behavioral modeling, can adequately capture both topological structure and behavioral semantics of RTL circuits.

**Text Modality Models**. We evaluate two leading text-based approaches: (1) **Qwen3-Embedding (Qwen3-E)** (Zhang et al., 2025), a state-of-the-art open-source embedding model with exceptional cross-lingual capabilities and strong performance across multiple natural language processing benchmarks; and (2) CodeV (Zhao et al., 2025), a specialized Verilog code understanding framework with three variants—**CodeV-CL-7B** (finetune based on CodeLlama-7b-Instruct (Roziere et al., 2023)), **CodeV-DS-6.7B** (finetune based on DeepSeek-Coder-6.7b-Instruct (Guo et al., 2024)), and **CodeV-QC-7B** (finetune based on Qwen2.5-Coder-7B (Hui et al., 2024)). Qwen3-E serves as a general-purpose text representation benchmark, while CodeV variants represent the current state-of-the-art in hardware-specific text modeling. These baselines collectively address the critical question of whether treating RTL as unstructured text (rather than recognizing its inherent graph structure) can effectively capture the essential characteristics of hardware designs, particularly the structured dataflow relationships that define circuit behavior.

**Multimodal Models**. **CircuitFusion** (Fang et al., 2025) represents the current frontier in multimodal circuit representation, integrating graph topology, natural language summaries, and raw RTL code through cross-modal attention mechanisms. Unlike TopoRTL, CircuitFusion relies on cross-stage netlist representations during pre-training to indirectly infer topological information, rather than explicitly modeling RTL's inherent graph structure. This baseline employs multiple contrastive learning objectives during the pretraining stage, including functional equivalence tasks. For a fair comparison, we remove the netlist encoder and only maintain the RTL encoder.

### C.5 Implementation Details

All experiments adhere to a rigorous implementation protocol designed to ensure fair, reproducible comparisons while maintaining fidelity to original methodologies.

**Graph Modality Models**. For GCN-based approaches, we implement a 3-layer GCN following the functional equivalence contrastive learning framework established in prior work (Xu et al., 2023). Functional equivalence pairs are systematically generated using Yosys for pre-training objectives. We maintain subgraph representation dimension at 768 across all graph models, with graph-level embeddings derived through sum-pooling operations. For GCN-GNN, we preserve the hierarchical graph processing approach with 3-layer GCN fine-tuning heads as in the original implementation. For GCN-MLP, we replace hierarchical processing with standard MLP heads to align with our unified evaluation framework. This dual-implementation strategy enables direct comparison between architecture-specific optimizations and standardized evaluation protocols.

**Text Modality Models**. For text-based approaches, we directly interface with Hugging Face APIs to obtain embeddings from Qwen3-Embedding and CodeV series models. Each circuit's representation is generated by concatenating the function description with corresponding RTL code, with truncation applied for sequences exceeding maximum token limits. Notably, we adopt different embedding extraction strategies aligned with each model's design philosophy: for CodeV variants, we use the mean of all hidden states in the final layer as the text embedding, while for Qwen3-Embedding, we utilize the last hidden state following its original implementation specifications. This approach ensures optimal utilization of each model's architectural strengths while maintaining consistent input processing across the text modality category.

**Multimodal Models**. For CircuitFusion implementation, we carefully follow its open-source code and published paper. The graph encoder employs a 7-layer Graphormer (Ying et al., 2021), producing 768-dimensional graph representations. The summary encoder utilizes the first 6 layers of BERT (Devlin et al., 2019) (768-dimensional hidden and output spaces), while the code encoder substitutes Qwen3-Embedding-0.6B for the originally proposed NV-Embd-V1 (Lee et al., 2024) due to hardware constraints on NVIDIA RTX 3090 GPUs. This substitution is justified by Qwen3-Embedding-0.6B's superior performance on the Massive Text Embedding Benchmark (MTEB) while maintaining the same 32K maximum input token capacity. Code embeddings (1024 dimensions) are linearly projected to 768 dimensions to maintain representation space consistency, with modality fusion handled by the final 6 layers of BERT.

**Evaluation Framework**. All models employ a standardized 768-dimensional output representation and undergo 50 pre-training epochs, with batch sizes adapted to graph scale: 128 for subgraph-based pre-training and 6 for full-graph pre-training. Text modality models are excluded from pre-training as their representations are directly obtained via API inference. Crucially, our evaluation protocol

extracts circuit representations at multiple pre-training epochs, with downstream task performance determining the optimal checkpoint selection based solely on validation set metrics. Dataset partitioning follows a 30%-30%-40% (train-validation-test) split at the circuit level, rather than subgraph level, to accommodate both global and subgraph-level downstream tasks while preventing data leakage. This partitioning strategy reflects real-world scenarios where substantial unlabeled data exists, emphasizing model generalization capabilities. All experiments were conducted on NVIDIA GeForce RTX 3090 GPUs.

# D  PRETRAINING AND EXPERIMENT RESULT DETAILS

## D.1  REGISTER CONE EXTRACT

---

**Algorithm 1** Register Cone Extraction via Register-Driven Backward Traversal

---

**Input:** RTL circuit $V$, Total registers $\{R_i\}_{i=1}^N$
**Output:** Register cones $\{V^{R_i}\}_{i=1}^N$
    **Phase 1: Build Signal Dependency Dictionaries**
1:   $D, C \leftarrow$ ParseVerilog($V$)    ▷ Extract signal declarations $D$ and combinational dependencies $C$
    **Phase 2: Backward Traversal from Registers**
2:   **for** each register $R_i \in \{R_i\}_{i=1}^N$ **do**
3:      $Q \leftarrow$ GetDrivingSignals($R_i$)                        ▷ Initialize with RHS signals of $R_i$
4:      $S \leftarrow \{R_i\}$, $I_{in} \leftarrow \{R_i\}$, $COI \leftarrow$ GetOutputDecl($R_i$)
5:      **while** $Q \neq \emptyset$ **do**
6:          $u \leftarrow Q$.dequeue()
7:          **if** $u \notin S$ **then**
8:             $S \leftarrow S \cup \{u\}$
9:             $I_{in} \leftarrow I_{in} \cup \{u\}$ if IsInputOrReg($u$)        ▷ Register-to-input conversion
10:         $COI \leftarrow COI \cup$ GetCodeLines($u, D, C$)    ▷ Add signal declaration/assignment
11:         $Q \leftarrow Q \cup$ ExtractDependencies($u, C$)    ▷ Backward propagate to RHS signals
                       ▷ Note: If $u$ is input/register, ExtractDependencies($u, C$) returns empty set
12:          **end if**
13:      **end while**
    **Phase 3: Sub-circuit Generation and Verification**
14:      $V^{R_i} \leftarrow$ GenerateModule($I_{in}, R_i, COI$)
15:      VerifyWithYosys($V^{R_i}$)                         ▷ Check for syntax correctness
16: **end for**
17: **return** $\{V^{R_i}\}_{i=1}^N$

---

**Algorithm Overview:** The register cone extraction process (Algorithm 1) systematically decomposes an RTL design into functionally complete subcircuits through three stages, ensuring both accuracy and scalability.

- **Phase 1: Build Signal Dependency Dictionaries**. Verilog code is parsed to extract two critical data structures: $D$, a dictionary mapping signals to their declarations; $C$, a directed graph encoding combinational dependencies between signals. These dictionaries enable precise tracking of signal origins and propagation paths, forming the foundation for subsequent traversal.

- **Phase 2: Backward Traversal from Registers**. For each register $R_i$, the algorithm initializes a queue $Q$ with its driving signals (RHS signals) and collects output declaration information $COI$. It then performs a backward traversal through combinational logic: starting from $R_i$, it dequeues signals $u$, adds them to the signal set $S$ if unvisited, and converts their input connections $I_{in}$ to corresponding declaration types (e.g., mapping register inputs to wire declarations). The traversal propagates upstream by enqueuing signals from $u$'s dependencies in $C$, recursively capturing all signals causally influencing $R_i$'s value, including indirect paths through intermediate registers. This phase ensures completeness by exhaustively tracing all upstream dependencies while avoiding redundant processing.

- **Phase 3: Sub-circuit Generation and Verification**. Using the collected signals $S$ and converted declarations, the algorithm generates a syntactically correct Verilog module $V^{R_i}$ for each register cone. This module includes: (1) all signals in $S$, (2) the original register $R_i$ and its driving combinatorial logic, and (3) corrected input declarations to ensure standalone functionality. To validate correctness, the generated subcircuit is verified using Yosys (Wolf et al., 2013), checking

for proper syntax, valid assignments, and resolved signal references. This step guarantees that each partitioned subcircuit is synthesizable and maintains behavioral integrity.

This framework achieves scalable and accurate decomposition by leveraging backward traversal to capture causal dependencies, ensuring completeness without over-inclusion. The integration of Yosys validation further enforces syntactic and functional correctness, making the approach robust for large-scale RTL designs.

### D.2 BEHAVIOR-AWARE TOKENIZERS PRETRAINING

#### D.2.1 GRAPH TOKENIZER

Graph Transformers have emerged as a powerful paradigm for modeling graph-structured data, directly addressing critical limitations of traditional message-passing GNNs, such as the over-smoothing problem. By replacing localized neighborhood aggregation with global attention mechanisms, Graph Transformers dynamically capture long-range dependencies while preserving structural uniqueness across all nodes. In this work, we adopt Graphormer (Ying et al., 2021) as our graph tokenizer to encode circuit topologies. Formally, given a sub-circuit $G^{R_i}$ with $N^{R_i}$ nodes, the output of tokenizer is:

$$x^{R_i},\ X^{R_i} = \text{Graph-Tokenizer}(G^{R_i}) \tag{16}$$

where $X^{R_i} \in \mathbb{R}^{N^{R_i} \times d}$ is the node feature matrix, and $x^{R_i} \in \mathbb{R}^{1 \times d}$ is a learnable [CLS] token to represent the global information.

**Behavior Equivalence Contrastive Learning.** To embed behavioral semantics into topology representations, we enforce that functionally equivalent circuits map to similar latent spaces. Given a sub-circuit $G^{R_i}$, we generate positive samples $G^{R_i}_{pos}$ using Yosys, which applies random structural transformations (e.g., gate resynthesis, buffer insertion) while preserving functional equivalence. Negative sample $G^{R_i}_{neg}$ is randomly selected from the same batch. We then optimize a contrastive loss using the TripletMarginLoss:

$$\mathcal{L}_{CL} = [\| x^{R_i} - x^{R_i}_{pos} \|_2^2 - \| x^{R_i} - x^{R_i}_{neg} \|_2^2 + \beta]_+, \tag{17}$$

where $\beta$ is a hyperparameter that controls the margin of the distance between pairs of positive and negative samples, and $[\cdot]_+$ is a shorthand for $\max(0, \cdot)$.

**Masked Node Modeling.** To help the model learn the topology connection relationships, we introduce a reconstruction task where random nodes are masked and their features predicted. For encoded node features $X^{R_i}$, we use a learnable [MASK] token to randomly mask nodes and obtain masked features $\tilde{X}^{R_i}$. The model then reconstructs the original features of masked nodes via a lightweight decoder head, optimized with mean squared error (MSE) loss:

$$\mathcal{L}_{mask} = -\frac{1}{|\mathcal{M}^{R_i}_G|} \sum_{j \in \mathcal{M}^{R_i}_G} \| \text{Decoder}(\tilde{X}^{R_i}_j) - X^{R_i}_j \|_2^2, \tag{18}$$

where $\mathcal{M}^{R_i}_G$ denotes the set of masked nodes. The total pretraining loss combines both objectives:

$$\mathcal{L}_{graph-tokenizer} = \lambda_1 \mathcal{L}_{CL} + \lambda_2 \mathcal{L}_{mask}, \tag{19}$$

where $\lambda_1, \lambda_2$ balance task contributions.

**Input Representation.** After pretraining the graph tokenizer, we initialize the representation using the [CLS] token in each sub-circuit and construct the input sequence for the entire design with a learnable global [CLS] token $x^{R_0}$:

$$X^0 = (x^{R_0 T}, x^{R_1 T}, \dots, x^{R_N T})^T \in \mathbb{R}^{(1+N) \times d} \tag{20}$$

#### D.2.2 SUMMARY TOKENIZER

Transformer-based language models have revolutionized natural language processing by effectively capturing contextual relationships through self-attention mechanisms. In this work, we adopt BERT (Devlin et al., 2019) as our summary tokenizer to encode textual descriptions of circuit behaviors. Formally, given a textual summary $S^{R_i}$ for sub-circuit $R_i$, the output of the tokenizer is:

$$t^{R_i},\ T^{R_i} = \text{Summary-Tokenizer}(S^{R_i}), \tag{21}$$

where $T^{R_i} \in \mathbb{R}^{TL^{R_i} \times d}$ is the token feature matrix with $TL^{R_i}$ representing the sequence length, and $t^{R_i} \in \mathbb{R}^{1 \times d}$ is the [CLS] token embedding that captures the global semantic representation of the summary.

**Behavior Equivalence Contrastive Learning.** To align textual representations with functional circuit semantics, we enforce that summaries describing functionally equivalent circuits map to similar regions in the embedding space. Given a sub-circuit $S^{R_i}$, we generate positive samples $S^{R_i}_{pos}$ by applying random but function-preserving transformations to the original circuit using Yosys, then re-generating the textual summary. Negative samples $S^{R_i}_{neg}$ are randomly selected from the same batch. We optimize the following contrastive loss using TripletMarginLoss:

$$\mathcal{L}_{CL} = [\| t^{R_i} - t^{R_i}_{pos} \|_2^2 - \| t^{R_i} - t^{R_i}_{neg} \|_2^2 + \beta]_+, \tag{22}$$

where $\beta$ is a hyperparameter that controls the margin of the distance between pairs of positive and negative samples, and $[\cdot]_+$ is a shorthand for $\max(0, \cdot)$.

**Masked Language Modeling.** To enhance the model's understanding of linguistic structure and circuit-specific terminology, we implement the standard BERT pretraining objective. After randomly masked tokens, the model then predicts the original tokens at masked positions through a classification head over the vocabulary. Formally, given masked token features $\tilde{T}^{R_i}$, the mask loss is computed as:

$$\mathcal{L}_{mlm} = -\frac{1}{|\mathcal{M}^{R_i}_S|} \sum_{j \in \mathcal{M}^{R_i}_S} \log p_\theta(T^{R_i}_j \mid \tilde{T}^{R_i}, \mathbf{A}^{R_i}), \tag{23}$$

where $\mathcal{M}^{R_i}_S$ denotes the set of masked token positions, $\mathbf{A}^{R_i}$ is the attention mask, and $p_\theta$ represents the probability distribution predicted by the model. The total pretraining loss combines both objectives:

$$\mathcal{L}_{summary-tokenizer} = \lambda_3 \mathcal{L}_{CL} + \lambda_4 \mathcal{L}_{mlm}, \tag{24}$$

where $\lambda_3, \lambda_4$ balance task contributions.

**Input Representation.** After pretraining the summary tokenizer, we extract the [CLS] token embedding from each summary to represent its semantic content. We then construct the input sequence for the entire design by concatenating these embeddings with a learnable global [CLS] token $t^{R_0}$:

$$T^0 = (t^{R_0 T}, t^{R_1 T}, \ldots, t^{R_N T})^T \in \mathbb{R}^{(1+N) \times d} \tag{25}$$

### D.3 DETAIL EXPERIMENT ANALYSIS

#### D.3.1 RQ1: PPA PREDICTION

To assess the ability to represent topology information, we performed five PPA prediction tasks focused on key metrics in circuit optimization. **Timing Performance**: Slack measures timing compliance post-synthesis, with Worst Negative Slack (WNS) indicating the largest timing violation, and Total Negative Slack (TNS) summing all violations to guide optimization efforts. **Area Performance**: Area refers to the total silicon area required for the circuit, crucial for feasibility and cost. **Power Performance**: Power measures the circuit's energy efficiency. Based on Tables 1, we can draw the following observations:

- **Obs: TopoRTL achieves superior area and power prediction with minimal resource overhead**. Specifically, it outperforms the best baseline by 5.5% ↑ in Area PCC and 6.9% ↑ in Power PCC, while slashing MAPE errors by 26.2% ↓ for area and 31.5% ↓ for power. Crucially, these improvements come with fewer parameters and training data, showcasing TopoRTL's effectiveness in capturing global topological dependencies that text-based models struggle with.
- **Obs: TopoRTL exhibits competitive timing performance due to its lightweight design**. It achieves the highest WNS prediction (PCC=0.862, RRSE=0.580), outperforming all baselines in critical-path topology modeling. Although it doesn't match the CodeV family for some timing tasks, it matches Slack PCC and surpasses most in RRSE, emphasizing the significance of topology-behavior integration. TopoRTL's WNS performance highlights its potential for timing optimization and scalability.
- **Obs: CodeV highlights domain-specific fine-tuning benefits but faces task-specific limitations**. It shows substantial gains over non-finetuned Qwen3, underscoring the critical role of

specialized training for RTL tasks. However, its improvements are constrained—e.g., it underperforms Qwen3-E-0.6B in WNS prediction and fails to achieve balanced results across all ppa tasks, revealing inherent limitations in model generalizability.

- **Obs: GCN-based models (GCN-MLP/GCN-GNN) exhibit poor accuracy due to topology-agnostic pretraining**. Their functional-aware contrastive learning discards essential circuit topology, as evidenced by identical graph representations for structurally distinct circuits (e.g., Circuit B vs. Circuit C in Figure 1). GCN-MLP typically outperforms GCN-GNN due to the invalid homophily assumption in circuit graphs and over-smoothing effects, while GCN-GNN excels in TNS tasks, indicating an edge in global analysis. Both models highlight that simple graph conversion isn't enough, emphasizing the need for topology-integrated designs in effective circuit modeling.

- **Obs: CircuitFusion's weak performance arises from topology information loss in architecture and pretraining**. CircuitFusion processes RTL code using functional contrastive pretraining focused on behavioral equivalence rather than topological relationships. While it converts RTL to CDFG representations, it fails to capture crucial topology-sensitive circuit information, as confirmed by GCN-GNN and GCN-MLP models. Our analysis shows that topology awareness depends on cross-stage netlist alignment; without this data, the model's topological awareness diminishes, degrading the reliability of timing predictions and underscoring the importance of topology.

Table 3: Detailed results of retrieval experiments.

| Method | AUC↑ | | | |
|---|---|---|---|---|
| | L=5 | L=8 | L=10 | L=15 |
| GCN-MLP | 0.684 | 0.630 | 0.504 | 0.499 |
| GCN-GNN | 0.664 | 0.507 | 0.500 | 0.500 |
| Qwen3-E-0.6B | 0.495 | 0.545 | 0.531 | 0.497 |
| Qwen3-E-4B | 0.489 | 0.512 | 0.505 | 0.500 |
| Qwen3-E-8B | 0.511 | 0.509 | 0.499 | 0.500 |
| CodeV-CL | 0.629 | 0.655 | 0.637 | 0.485 |
| CodeV-DS | 0.551 | 0.523 | 0.572 | 0.631 |
| CodeV-QC | 0.522 | 0.530 | 0.509 | 0.509 |
| CircuitFusion | 0.674 | 0.674 | 0.666 | 0.670 |
| TopoRTL | 0.787 | 0.804 | 0.760 | 0.783 |

### D.3.2   RQ2: CIRCUIT RETRIEVE

To evaluate behavioral representation capabilities, we conduct a natural language code search task critical for hardware design reuse and verification. Following Lu et al. (2021), we evaluate with $L$ negative designs ($L \in \{5, 8, 10, 15\}$) per query, measuring performance via AUC. Further details regarding this task can be found in Appendix C.2. Based on Figure 3 and Table 3, we derive the key insights below:

- **Obs: TopoRTL demonstrates superior performance and robustness across retrieval scenarios**. Our model maintains a stable performance near 0.8 AUC for all $L$ values (5-15 negative samples), outperforming all baselines. This consistency stems from TopoRTL's joint modeling of topology and behavior, emphasizing the importance of topology in RTL representation learning. The topology-guided alignment mechanism filters out irrelevant samples, ensuring reliable behavioral matching even in noisy conditions, thus enhancing cross-modal retrieval accuracy and supporting scalable design reuse across various hardware applications.

- **Obs: CodeV validates domain adaptation efficacy in retrieval tasks through consistent gains over Qwen3**. It achieves a higher AUC than non-finetuned Qwen3 across all negative sample lengths (5–15 negative samples), demonstrating that RTL-specific fine-tuning effectively captures behavioral semantics for retrieval. This reinforces domain adaptation as a critical strategy, though its task-specific limitations persist.

- **Obs: GCN models approach behavioral retrieval performance but reveal topology's critical role**. While GCN models generally trail behind CircuitFusion, they surpass LLM-based models in retrieval AUC, confirming that graph modality with functional contrastive learning effectively captures behavioral semantics. However, their consistent underperformance against TopoRTL

demonstrates that behavioral modeling alone is insufficient; precise topological integration remains indispensable for robust cross-modal retrieval.

### D.3.3 RQ3: HIDDEN REPRESENTATION ANALYSIS

As demonstrated in the previous sections, TopoRTL effectively learns both topological and behavioral circuit characteristics. To further validate this, we visualize the learned representations using t-SNE (Maaten & Hinton, 2008). Embeddings from our model and CircuitFusion (selected as it matches TopoRTL's output dimension and training data scale) are projected into 2D space, colored by normalized Area, Power, and Slack metrics. According to Figure 4, we can find that:

- **Obs: TopoRTL preserves continuous topological trends in representation space**. In *Area* and *Power* visualizations (Figure 4b), TopoRTL exhibits smooth, coherent gradients along t-SNE dimensions, evidenced by seamless purple-to-yellow shifts for Area and Power. This reflects precise modeling of topological scaling effects (e.g., larger circuits systematically mapping to higher Area/Power regions). Conversely, CircuitFusion (Figure 4a) shows fragmented, discontinuous distributions with abrupt value jumps (e.g., isolated high-Power clusters amid low-Power regions), indicating failure to capture topological continuity. This validates TopoRTL's topology-guided alignment in preserving quantitative design variations.
- **Obs: TopoRTL achieves topology-aware clustering for discrete design regimes**. For *Slack* prediction (Figure 4b), TopoRTL forms distinct, non-overlapping clusters: high-Slack circuits (orange/yellow) cleanly separate from low-Slack regions (blue/purple), directly corresponding to critical-path topologies. CircuitFusion (Figure 4a) exhibits severe cluster entanglement, proving its inability to disentangle topologically critical states. This confirms TopoRTL's unique capacity to encode discrete topological regimes essential for timing-critical decision making, a capability absent in behavior-only models.
- **Obs: Discrete representation gaps in TopoRTL hint at RTL-to-gate-level topological mismatches**. While TopoRTL successfully clusters Slack values, isolated outliers (e.g., yellow points) suggest unresolved discrepancies between abstract RTL descriptions and concrete gate-level implementations. These gaps likely stem from: (1) Abstraction loss: RTL netlists omit low-level details (e.g., buffer insertion, wire routing) critical for precise timing analysis; (2) Hierarchical misalignment: Modular RTL components may map non-linearly to flat gate-level structures, disrupting topological continuity. This observation highlights the necessity for improved and well-designed topology features.

### D.3.4 RQ4: ABLATION AND FURTHER ANALYSIS

**Abalation Study**. To rigorously validate the contribution of each TopoRTL component, we conduct ablation experiments by systematically removing key modules: (1) w/o Bit-width: eliminating bit-width centrality encoding $a^{bit}$ and feeding initial embeddings directly to the transformer; (2) w/o Max-path: discarding max-path discrepancy encoding $\Delta L$ during attention score computation; (3) w/o Graph density: removing graph density encoding $\Delta\rho$ from attention mechanisms; (4) w/o Cross-loss: replacing topology-guided alignment with standard contrastive learning between isolated graph and text modalities. As shown in Figure 5 and Table 4, these experiments reveal:

- **Obs: Positional encodings yield balanced performance across diverse downstream tasks**. Bit-width centrality encoding improves performance by effectively capturing both topology and functional complexity. In contrast, max-path and density encodings demonstrate varying impacts due to gaps between RTL and netlist representations. This suggests that a comprehensive representation of a circuit requires complementary topological signals.
- **Obs: Topology-guided cross-modal alignment prioritizes topology fidelity over pure timing accuracy**. By enforcing topology-semantic consistency, the alignment ensures behavioral descriptions honor physical constraints, which is a necessary trade-off for design left-shift that slightly constrains timing prediction (e.g., TNS) while significantly boosting other topological and behavioral tasks. This confirms that topology-guided alignment is helpful for end-to-end design optimization.

**Effect of Dataset Scale**. As noted earlier, TopoRTL slightly underperforms larger models on timing tasks due to its small pretraining dataset. Given EDA's scarcity of labeled data, evaluating low-label generalization is critical for real-world deployment. To assess scalability and data efficiency, we

Table 4: Detailed results of the ablation study

| Model | Area | | | | Power | | | | Slack | | | |
|---|---|---|---|---|---|---|---|---|---|---|---|---|
| | PCC↑ | $R^2$↑ | MAPE↓ | RRSE↓ | PCC↑ | $R^2$↑ | MAPE↓ | RRSE↓ | PCC↑ | $R^2$↑ | MAPE↓ | RRSE↓ |
| TopoRTL | 0.863 | 0.683 | 7.952 | 0.574 | 0.884 | 0.712 | 25.033 | 0.585 | 0.909 | 0.821 | 31.249 | 0.443 |
| w/o cross modal loss | 0.839 | 0.662 | 8.992 | 0.602 | 0.859 | 0.695 | 28.098 | 0.636 | 0.892 | 0.792 | 32.720 | 0.491 |
| w/o graph density | 0.851 | 0.692 | 8.794 | 0.572 | 0.874 | 0.689 | 29.777 | 0.621 | 0.873 | 0.744 | 38.662 | 0.536 |
| w/o max path | 0.854 | 0.705 | 8.634 | 0.565 | 0.871 | 0.694 | 27.998 | 0.616 | 0.890 | 0.777 | 33.599 | 0.525 |
| w/o bit width | 0.838 | 0.693 | 9.354 | 0.645 | 0.792 | 0.553 | 34.159 | 0.755 | 0.854 | 0.709 | 32.479 | 0.707 |
| only graph density | 0.836 | 0.696 | 9.652 | 0.632 | 0.812 | 0.627 | 37.818 | 0.649 | 0.837 | 0.690 | 35.348 | 0.717 |
| only max path | 0.816 | 0.660 | 10.085 | 0.730 | 0.831 | 0.637 | 32.198 | 0.645 | 0.838 | 0.605 | 40.505 | 0.664 |
| only bit width | 0.835 | 0.663 | 8.889 | 0.614 | 0.814 | 0.611 | 29.139 | 0.669 | 0.856 | 0.717 | 39.541 | 0.557 |

| Model | TNS | | | | WNS | | | | Retrieval |
|---|---|---|---|---|---|---|---|---|---|
| | PCC↑ | $R^2$↑ | MAPE↓ | RRSE↓ | PCC↑ | $R^2$↑ | MAPE↓ | RRSE↓ | AUC↑ |
| TopoRTL | 0.872 | 0.743 | 32.016 | 0.521 | 0.862 | 0.723 | 40.130 | 0.580 | 0.787 |
| w/o cross modal loss | 0.902 | 0.800 | 31.776 | 0.444 | 0.869 | 0.710 | 35.936 | 0.633 | 0.759 |
| w/o graph density | 0.867 | 0.723 | 32.821 | 0.515 | 0.867 | 0.734 | 41.962 | 0.555 | 0.771 |
| w/o max path | 0.901 | 0.778 | 32.937 | 0.503 | 0.896 | 0.760 | 41.148 | 0.564 | 0.781 |
| w/o bit width | 0.882 | 0.722 | 37.991 | 0.601 | 0.813 | 0.545 | 41.096 | 0.913 | 0.723 |
| only graph density | 0.893 | 0.762 | 36.002 | 0.504 | 0.855 | 0.642 | 42.884 | 0.840 | 0.709 |
| only max path | 0.858 | 0.663 | 34.672 | 0.584 | 0.726 | 0.265 | 55.031 | 0.988 | 0.684 |
| only bit width | 0.896 | 0.773 | 33.819 | 0.446 | 0.867 | 0.739 | 44.679 | 0.526 | 0.760 |

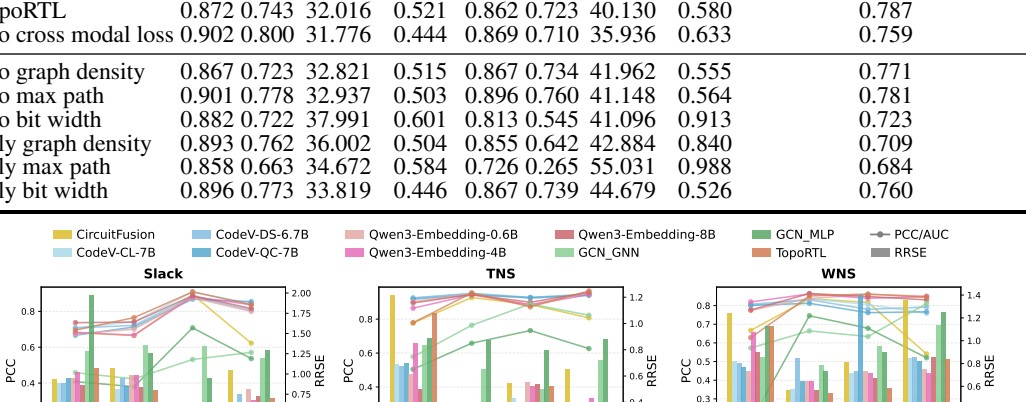

Figure 7: Split Ratio Results.

train TopoRTL at 10%, 20%, 30% (default), and 40% label rates, using equal validation splits with the remainder as test data. As shown in Figure 7, results reveal:

- **Obs: TopoRTL surpasses larger models across more tasks at sufficient label rates**. The model initially underperforms baselines at lower label rates (10% and 20%), but consistently surpasses or at least matches all competing approaches when label rates reach 30% and 40%. This progression confirms that TopoRTL's topology-aware architecture efficiently learning the topology and behavior information, proving its viability for industrial EDA pipelines where labeled data gradually accumulates.

# E FURTHER ANALYSIS: ROBUSTNESS AND GENERALIZATION

## E.1 CIRCUIT SCALE ANALYSIS

Real-world applications involve diverse circuits with varying functionalities and sizes, necessitating representation models that possess both robustness and scalability. To further investigate model

Table 5: Performance comparison across different circuit scales. (MAPE%)

| Method | Task | Small | Medium | Large | Mean | Std |
|---|---|---|---|---|---|---|
| TopoRTL | Area | **8.178** | **6.231** | **11.924** | **8.778** | 2.893 |
| | Power | **28.4** | **18.846** | 29.094 | **25.447** | **5.727** |
| | Slack | 62.187 | 37.242 | 33.587 | 44.339 | **15.565** |
| | TNS | **45.243** | 15.834 | 24.954 | 28.677 | **15.054** |
| | WNS | 53.362 | **25.937** | 27.414 | 35.571 | 15.425 |
| CircuitFusion | Area | 14.001 | 8.273 | 24.285 | 15.520 | 8.113 |
| | Power | 54.046 | 24.835 | 50.857 | 43.246 | 16.024 |
| | Slack | **59.419** | 36.396 | **25.491** | **40.435** | 17.321 |
| | TNS | 50.146 | **12.485** | 33.930 | 32.187 | 18.891 |
| | WNS | **46.580** | 30.336 | **27.172** | **34.696** | **10.413** |
| CodeV-DS | Area | 11.818 | 8.504 | 13.058 | 11.127 | **2.354** |
| | Power | 48.724 | 22.734 | **26.950** | 32.803 | 13.948 |
| | Slack | 63.297 | **32.643** | 30.260 | 42.067 | 18.425 |
| | TNS | 46.043 | 18.053 | **14.222** | **26.106** | 17.372 |
| | WNS | 52.151 | 31.809 | 28.311 | 37.424 | 12.874 |

performance across different scales, we partitioned the test set based on post-synthesis logic cell counts into three categories: *Small* ($< 1k$ cells), *Medium* (1k-10k cells), and *Large* ($> 10k$ cells). We selected three representative top-performing models for comparison: CodeV-DS (text-based), CircuitFusion (multimodal), and TopoRTL. As shown in Table 5, the results reveal two key insights:

- **Obs: TopoRTL demonstrates superior robustness on large-scale designs.** On topology-sensitive metrics such as Area and Power, TopoRTL maintains high accuracy even as circuit complexity increases. Notably, on *Large* circuits, TopoRTL achieves an Area MAPE of $11.92\%$, reducing the error by nearly $50\%$ compared to CircuitFusion ($24.29\%$). This confirms that our explicit topology encodings effectively capture the complexity of large-scale combinational logic blocks.
- **Obs: TopoRTL exhibits large scale invariance.** While baseline models often show significant performance fluctuation across different groups, TopoRTL maintains consistent low variance across Small, Medium, and Large categories. This suggests that our model generalizes well to unseen designs across different scales.

### E.2 CROSS-PDK GENERALIZATION ANALYSIS

During logic synthesis, RTL circuits are mapped to physical gates based on a specific Process Design Kit (PDK). Consequently, PPA metrics derived from different manufacturing processes (e.g., varying nanometer nodes) exhibit significant discrepancies. However, since RTL descriptions fundamentally specify functional behavior and logical topology rather than physical implementation, the representations learned by TopoRTL should theoretically be PDK-agnostic, enabling early-stage optimization across diverse technologies.

To validate this cross-PDK generalization, we re-synthesized the dataset using two additional open-source PDKs, SkyWater 130nm and GlobalFoundries 180nm, distinct from the default NanGate 45nm. As illustrated in Figure 8, PPA distributions vary significantly across process nodes, with performance metrics naturally degrading as the node size increases. To address absolute scale differences while preserving relative circuit rankings, we applied a log-transformation to the PPA labels. We compared TopoRTL against CodeV-DS and CircuitFusion, with results summarized in Table 6. The analysis yields two key observations:

- **Obs: TopoRTL achieves robust generalization across technologies.** TopoRTL consistently outperforms baselines on 45nm, 130nm, and 180nm tasks when prediction heads are trained on the corresponding data. For instance, in Area prediction on GlobalFoundries 180nm, TopoRTL achieves a PCC of $0.808$, significantly surpassing CircuitFusion ($0.620$) and CodeV-DS ($0.791$). This confirms that our model learns universal circuit properties.
- **Obs: Performance trends.** All models show slightly reduced accuracy on older PDKs (130nm/180nm vs. 45nm), primarily because open-source PDKs have limited standard cell li-

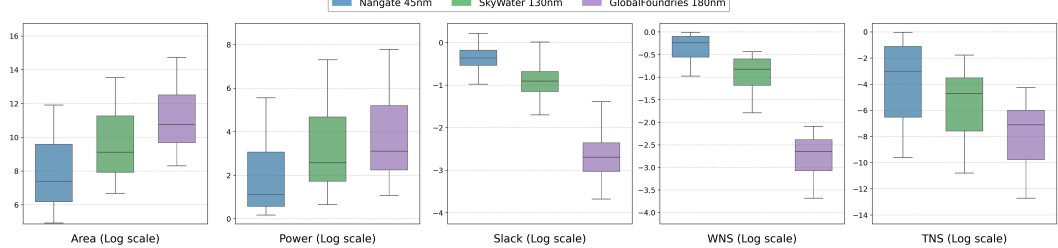

Figure 8: Label distribution statistics on different PDKs.

Table 6: Performance comparison on different PDKs.

| Model | Task | NanGate 45nm | | SkyWater 130nm | | GlobalFoundries 180nm | |
|---|---|---|---|---|---|---|---|
| | | PCC↑ | MAPE↓ | PCC↑ | MAPE↓ | PCC↑ | MAPE↓ |
| TopoRTL | Area | **0.863** | **7.952** | **0.833** | **7.351** | **0.808** | **6.313** |
| | Power | **0.884** | **25.033** | **0.896** | **16.624** | **0.873** | **14.649** |
| | Slack | **0.909** | 31.249 | 0.876 | 26.957 | 0.830 | 11.025 |
| | TNS | 0.872 | 32.016 | 0.901 | **13.036** | 0.902 | **7.849** |
| | WNS | **0.862** | 40.130 | **0.820** | **17.890** | 0.806 | 8.553 |
| CircuitFusion | Area | 0.647 | 13.242 | 0.643 | 10.804 | 0.620 | 8.664 |
| | Power | 0.657 | 43.073 | 0.643 | 34.308 | 0.640 | 28.268 |
| | Slack | 0.893 | **30.944** | 0.885 | 25.408 | 0.865 | 9.451 |
| | TNS | 0.885 | 34.454 | 0.848 | 18.315 | 0.831 | 11.983 |
| | WNS | 0.817 | **38.227** | 0.798 | 21.267 | **0.852** | 7.399 |
| CodeV-DS | Area | 0.814 | 10.778 | 0.806 | 8.989 | 0.791 | 7.397 |
| | Power | 0.827 | 36.544 | 0.818 | 27.850 | 0.819 | 21.950 |
| | Slack | 0.881 | 32.712 | **0.900** | **22.761** | 0.905 | 7.325 |
| | TNS | **0.928** | **31.857** | **0.918** | 15.977 | 0.917 | 9.397 |
| | WNS | 0.780 | 41.750 | 0.763 | 17.954 | 0.790 | **7.330** |

braries, causing synthesis tools to map complex functions to suboptimal cells and thus introducing noise in ground-truth PPA labels.

### E.3    ROBUSTNESS ABILITY ON SUMMARY NOISE

In TopoRTL, we leverage dual modalities, graph and summary, to capture structural topology and behavioral semantics, respectively. For the summary modality, we employ LLMs to generate functional descriptions from Verilog code, which are then processed by a behavior-aware tokenizer. However, given the varying capabilities of different LLMs or human-written types, the quality of generated text can fluctuate. To investigate the sensitivity of RTL representation learning to text quality, we conducted experiments from two perspectives: semantic quality variation and extreme information mismatch.

**Impact of Semantic Quality.** We first simulated a scenario with lower-quality textual inputs by replacing the summaries generated by GPT-OSS-120B with those from a significantly smaller model, GPT-OSS-20B. The TopoRTL model was then retrained using these coarser summaries. As illustrated in Figure 9, the results reveal:

• **Obs: TopoRTL is robust to variations in LLM capability.** The performance degradation across all downstream tasks remains minimal (PCC, $< 4\%$) when switching to the smaller 20B model. This indicates that while high-fidelity summaries optimize performance, TopoRTL does not strictly depend on state-of-the-art LLMs. The graph modality acts as a structural anchor, stabilizing the learned representation even when textual nuances are less precise, ensuring broad applicability even with resource-constrained generation models.

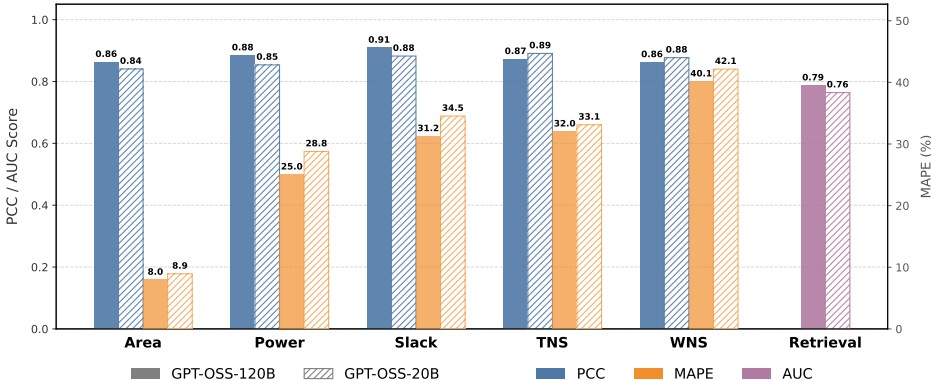

Figure 9: Performance of using different LLMs to generate functionality summary.

Table 7: Performance comparison across different summary shuffling ratios.

| Task | 0% | | 1% | | 5% | | 10% | |
|---|---|---|---|---|---|---|---|---|
| | PCC↑ | MAPE↓ | PCC↑ | MAPE↓ | PCC↑ | MAPE↓ | PCC↑ | MAPE↓ |
| Area | 0.8629 | 7.9521 | 0.869 | 7.7672 | 0.8421 | 9.0629 | 0.8057 | 9.5074 |
| Power | 0.8842 | 25.0326 | 0.8315 | 28.6588 | 0.8196 | 31.9563 | 0.8384 | 31.1551 |
| Slack | 0.9089 | 31.2487 | 0.8778 | 34.2507 | 0.8695 | 39.4997 | 0.8795 | 34.1435 |
| TNS | 0.8723 | 32.0156 | 0.8798 | 33.1328 | 0.8666 | 34.0015 | 0.8734 | 34.0001 |
| WNS | 0.8621 | 40.1298 | 0.8814 | 43.4544 | 0.8851 | 37.3168 | 0.8926 | 39.2374 |
| Retrieval | 0.787 | | 0.7705 | | 0.7651 | | 0.7675 | |

**Impact of Textual Accuracy.** We further introduced extreme noise to test the model's ability to correct misinformation. During pretraining, we randomly shuffled the textual summaries for a specific ratio (0%, 1%, 5%, 10%) of the dataset (effectively pairing circuits with completely incorrect descriptions), forcing the model to reconcile conflicting modal signals. As shown in Table 7:

- **Obs: Topology encodings act as a correction mechanism for behavioral noise.** We observed that timing-related tasks remain highly resilient to textual noise. Remarkably, WNS prediction performance actually improves under high noise conditions. We attribute this to the intrinsic conflict between high-level summaries and worst-case timing. Text summaries provide a prior for "average" functional behavior but lack specific information about the critical path. When textual inputs are noisy, the model effectively gates out these vague semantic signals and is forced to rely exclusively on the precise, explicitly encoded topology. This confirms that our topology-aware architecture provides a robust backbone that compensates when behavioral descriptions fail. Small circuits have few registers (<5), making global property (Area/Power) estimation vulnerable to semantic corruption.

### E.4 Circuit Functionality and Topology Similarity Analysis

From a hardware perspective, an RTL circuit is inherently a structured dataflow graph where behavioral intent and topological structure are intimately bound. As illustrated in Figure 1, circuits can share identical functions (e.g., Circuit B and C) yet exhibit divergent topologies that dictate physical performance (PPA). Therefore, an ideal RTL representation must simultaneously capture behavioral equivalence while distinguishing topological variations.

To further validate TopoRTL's capability in disentangling these aspects, we employed Yosys to generate functionally equivalent but structurally diverse variants for each circuit in our dataset. For every original-variant pair, we extracted both global embeddings (derived from the [CLS] token) and subgraph embeddings (register cone level), and computed their cosine similarities. The resulting distributions are presented in Figure 10. The analysis yields two critical observations:

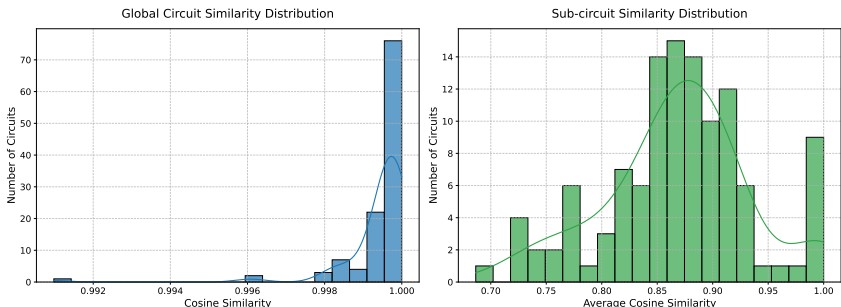

Figure 10: Cosine similarity histogram of functionally equivalent circuits.

- **Obs: TopoRTL achieves near-perfect behavioral consistency.** The global embeddings of functionally equivalent pairs exhibit an extremely high mean similarity of $0.999$. This confirms that our behavior-aware dual-modal tokenizers successfully align the high-level semantic representation of circuits, ensuring that structural transformations do not distort the model's understanding of the underlying functionality.
- **Obs: TopoRTL maintains acute topological sensitivity within functional equivalence.** In contrast to the global alignment, the subgraph embeddings show a noticeably lower mean similarity of $0.868$. This distinct "similarity gap" proves that our topology-aware encodings and alignment effectively detect and encode local structural variations (such as logic depth changes or interconnection density) even when the overall function remains unchanged. This capability is pivotal for precise PPA prediction, as it allows the model to differentiate between implementation choices that affect timing and power without losing functional context.

This directly addresses our core question from the introduction: TopoRTL uniquely balances behavioral equivalence preservation with topological differentiation.

## F FURTHER DISCUSSION AND FUTURE OUTLOOK

While TopoRTL successfully demonstrates the importance of topology in RTL representation learning for PPA prediction and retrieval, our framework opens up several promising directions for future research and industrial applications.

**Application to Functional Verification Tasks.** Current evaluations focus on static analysis (PPA and Retrieval). However, the principles of TopoRTL are theoretically well-suited for dynamic functional verification, such as coverage prediction (e.g., Design2Vec Vasudevan et al. (2021)). Verification is fundamentally a problem of state reachability and logic dependency. Our *Bit-Width Centrality Encoding* inherently captures state space dimensionality, while the *Register Cone* decomposition mirrors the "Cone of Influence" analysis used in formal verification. Future work will explore leveraging these topological priors to predict verification complexity and guide testbench generation, bridging the gap between static representation and dynamic behavior.

**Generalizing to Gate-Level Representations.** The core insight of TopoRTL, explicitly modeling the interplay between topological structure and functional behavior, is not limited to RTL but is transferrable to lower levels of abstraction, such as gate-level netlists. The concept of register cones naturally extends to Flip-Flop (DFF) cones in netlists. At this level, our positional encodings become even more physically meaningful: *Max-Path* maps directly to critical path delays through standard cells, and *Graph Density* correlates strongly with routing congestion. Adapting TopoRTL to netlists could enable fine-grained physical design prediction, creating a unified representation learning framework across the design flow.

**Towards Real-Time EDA Integration.** Unlike large-scale LLMs that suffer from high latency, TopoRTL is lightweight (29.13M parameters) and efficient, with an average inference time of less than one second per circuit. This efficiency, combined with our superior accuracy in PPA prediction, makes TopoRTL an ideal candidate for integration into real-time EDA flows. We envision TopoRTL functioning as an interactive "copilot" within design tools, providing instant feedback on power and timing implications as engineers modify code, thereby accelerating the iterative loop of agile chip design.

## G  OTHER DETAILS

---

**Prompts to generate module-level and design-level descriptions**

### Module-level Generation

| | |
|---|---|
| System Prompt | You are a professional VLSI designer and an expert at Verilog coding. Your task is to analyze a Verilog module and provide a structured description in JSON format. |
| User Prompt | Analyze the following Verilog module. Your response MUST be a single, valid JSON object. 
 Do not include any introductory text or explanations outside of the JSON structure. 
 The JSON object should have the following keys: 
 1. "suggested_name": A short, descriptive, and functional name for the module (e.g., "ALU", "FIFO Controller"). 
 2. "inputs": A list of strings, where each string is a high-level description of an inputś purpose (e.g., "Clock signal", "Data to be written", "Reset signal"). Do not use signal names from the code. 
 3. "outputs": A list of strings, similar to inputs, describing each outputś purpose (e.g., "Result of calculation", "Indicates buffer is full"). 
 4. "functionality": A concise paragraph describing what the module does, its main operations, and its purpose. Avoid implementation details. 
 5. "sub_modules_called": A list of strings containing the names of any other modules instantiated within this module. If none, provide an empty list []. 

 Here is the Verilog module code: 
 ```verilog 
 {module_code} 
 ``` |

### Design-level Generation

| | |
|---|---|
| System Prompt | You are a professional VLSI designer and an expert technical writer. You synthesize descriptions of individual circuit modules into a cohesive, high-level overview of the entire design. |
| User Prompt | You are given descriptions for individual hardware modules that make up a larger digital circuit. Your task is to generate a single, high-level natural language description of the **entire circuit's functionality**. Follow these requirements: 
 1. Focus on the overall purpose and main operations of the complete design. Synthesize, do not just list the parts. 
 2. Do not include any variable names, signal names, or the suggested module names from the provided context. 
 3. The description should be concise, clear, and written as if a human user is describing what they want the final circuit to achieve. 
 4. Keep the final description under 400 words. 

 Here are the descriptions of the individual modules: 

 {context_str} |

