# OpenReview forum: "Topology Matters in RTL Circuit Representation Learning"
_ICLR.cc/2026/Conference — ICLR 2026 Poster_

### Official Review · Reviewer_e2DU · 2025-10-27

**Soundness:** 3
**Presentation:** 3
**Contribution:** 3
**Rating:** 6
**Confidence:** 4

**Summary:**

This paper introduces TopoRTL, a novel framework for representation learning of RTL circuits that explicitly models both behavioral semantics and topological structures. Existing LLM-based or text-only RTL methods (e.g., CodeV, DeepRTL) treat Verilog code as pure text, overlooking the inherent dataflow topology of circuits. TopoRTL addresses this limitation through three key components: behavior-aware dual-modal tokenizers, which capturing both graph-based (structural) and textual (behavioral) modalities; topology-aware positional encodings, which integrating bit-width, path length, and graph density into Transformer attention to represent structural hierarchy and timing characteristics; topology-guided cross-modal alignment, which ensuring semantic consistency between the two modalities under topological constraints. Experiments on PPA prediction and natural language circuit retrieval show substantial gains over prior baselines such as CodeV, Qwen3, and CircuitFusion, with fewer parameters. The results confirm that explicit topology modeling is crucial for RTL representation learning.

**Strengths:**

1.	The paper introduces a novel perspective on RTL representation learning by emphasizing the importance of topological structure in addition to behavioral semantics. The idea of integrating bit-width, signal-path, and graph-density encodings into Transformer attention is new in the RTL/EDA domain. Furthermore, the cross-modal alignment strategy between topology and textual modalities represents a creative extension of multimodal learning techniques into hardware design contexts.
2.	The methodological pipeline—from register cone extraction to dual-modal embedding and topology-guided alignment—is technically sound and carefully justified. Experiments are extensive and clearly show that TopoRTL achieves consistent improvements across PPA prediction and circuit retrieval benchmarks, outperforming both large-scale LLMs (e.g., CodeV, Qwen3) and graph-based methods.
3.	The paper is well organized, with a clear motivation and coherent narrative flow. Figures (especially Fig. 1–3) effectively illustrate the problem and model design. The appendices provide detailed implementation and dataset descriptions, enhancing reproducibility.
4.	This work bridges two rapidly growing research areas—representation learning and EDA automation—demonstrating that explicitly modeling topology leads to superior circuit representations. The framework could meaningfully impact early-stage chip design optimization and inspire future work on structure-aware learning across code and graph modalities.

**Weaknesses:**

1.	The t-SNE visualization (Figure 4) shows separation but doesn’t analyze how topology encoding affects cluster formation. A more explicit comparison of embedding similarities (e.g., through cosine distances between functionally equivalent circuits) could strengthen the argument.
2.	The summary modality relies on GPT-OSS-120B to generate textual circuit descriptions. The paper doesn’t analyze how summary quality affects downstream performance or whether the model could generalize with human-written or noisy summaries.

**Questions:**

1.	During training, how sensitive is the model to the quality of text summaries? If summaries are partially inaccurate, does the topology encoder still preserve robustness?
2.	Could the proposed topology-aware encoding generalize beyond RTL (e.g., to gate-level netlists or analog blocks)?
3.	What is the quantitative contribution of each topology encoding (bit-width, path, density) when combined pairwise, rather than removed individually?
4.	Could TopoRTL be integrated into synthesis or verification tools for real-time PPA estimation?

---

> ### Author Response · Authors · 2025-11-25
> **Response to Reviewer e2DU (1/5)**
>
> We thank the reviewer for recognizing our novel perspective on RTL representation learning through explicit topology modeling and our technically sound methodology that achieves consistent improvements over both large-scale LLMs and graph-based methods.
> We have comprehensively addressed these concerns regarding topology encoding visualization (W1) through functional equivalence analysis demonstrating behavioral preservation and topological sensitivity (A1),
> validated robustness to summary quality variations (W2&Q1) via controlled experiments with lower-quality and noisy summaries (A2),
> explored broader applicability of our core principle to gate-level netlists (Q2) (A3),
> and quantified the complementary contributions of each topology encoding through pairwise and single-encoding ablation studies (Q3) (A4).
>
> > W1: The t-SNE visualization (Figure 4) shows separation but doesn’t analyze how topology encoding affects cluster formation. A more explicit comparison of embedding similarities (e.g., through cosine distances between functionally equivalent circuits) could strengthen the argument.
>
> **A1**: We thank the reviewer for this crucial suggestion.
> While Figure 4 demonstrates topology-PPA alignment through smooth gradients in t-SNE visualizations,
> we now provide more evidence of how topology encoding shapes embeddings using functionally equivalent circuits.
>
> We use Yosys with standard optimization passes (e.g., opt_expr, opt_merge) to generate structurally diverse but functionally equivalent variants for our dataset.
> For each original-variant pair:
> * We extracted global circuit embeddings (from [CLS] token) and subgraph embeddings (register-cone level)
> * Computed cosine similarity between embeddings of functionally equivalent pairs
>
> Results:
>
> | Representation                | Mean  | Median | Std   |
> |-------------------------------|-------|--------|-------|
> | Global embedding similarity   | 0.999 | 0.999  | 0.001 |
> | Subgraph embedding similarity | 0.868 | 0.872  | 0.066 |
>
>
> Key insights:
> * **Behavioral Preservation**: TopoRTL achieves near-perfect global similarity (0.999) for functionally equivalent circuits, confirming its ability to preserve behavioral semantics despite structural variations.
> * **Topological Sensitivity**: Crucially, TopoRTL shows lower subgraph similarity, proving our topology encodings and alignment detect structural differences within functionally equivalent designs.
>
> This directly addresses our core question from the introduction: TopoRTL uniquely balances behavioral equivalence preservation with topological differentiation.

---

> ### Author Response · Authors · 2025-11-25
> **Response to Reviewer e2DU (2/5)**
>
> > W2&Q1: The summary modality relies on GPT-OSS-120B to generate textual circuit descriptions. The paper doesn’t analyze how summary quality affects downstream performance or whether the model could generalize with human-written or noisy summaries. During training, how sensitive is the model to the quality of text summaries? If summaries are partially inaccurate, does the topology encoder still preserve robustness?
>
> **A2**: We thank the reviewer for this critical question.
> While we use GPT-OSS-120B for summary generation to capture precise behavioral semantics, we rigorously evaluate sensitivity to summary quality through two experiments:
>
> 1. Lower-quality summaries
>
> During training, we replace GPT-OSS-120B summary with GPT-OSS-20B summary, and retrain TopoRTL. Results as below:
>
> | Task           | GPT-OSS-120B |        | GPT-OSS-20B |        |
> |----------------|--------------|--------|-------------|--------|
> |                | PCC          | MAPE   | PCC         | MAPE   |
> | Area           | 0.863        | 7.952  | 0.841       | 8.939  |
> | Power          | 0.884        | 25.033 | 0.854       | 28.789 |
> | Slack          | 0.909        | 31.249 | 0.882       | 34.543 |
> | TNS            | 0.872        | 32.016 | 0.891       | 33.118 |
> | WNS            | 0.862        | 40.130 | 0.877       | 42.149 |
> | Retrieve (AUC) | 0.787        |        | 0.764       |        |
>
> Performance **degradation remains <4%** across PPA tasks (PCC) and the Retrieve task (AUC).
> This confirms that high-quality summaries optimize performance.
> Meanwhile, modern LLMs (e.g., GPT-5, Gemini 3.0) now consistently exceed GPT-OSS-120B’s capabilities, making high-fidelity summaries increasingly accessible.
>
> 2. Extreme noise injection (random summary shuffling):
>
> During training, we randomly shuffled summaries for x% of circuits (creating completely incorrect descriptions), then evaluated with clean summaries:
>
> | Task           | 0%    |        | 1%    |        | 5%    |         | 10%    |        |
> |----------------|-------|--------|-------|--------|-------|---------|--------|--------|
> |                | PCC   | MAPE   | PCC   | MAPE   | PCC   | MAPE    | PCC    | MAPE   |
> | Area           | 0.863 | 7.952  | 0.869 | 7.767  | 0.842 | 9.063   | 0.806  | 9.507  |
> | Power          | 0.884 | 25.033 | 0.832 | 28.659 | 0.820 | 31.956  | 0.838  | 31.155 |
> | Slack          | 0.909 | 31.249 | 0.878 | 34.251 | 0.870 | 39.500  | 0.880  | 34.144 |
> | TNS            | 0.872 | 32.016 | 0.880 | 33.133 | 0.867 | 34.002  | 0.873  | 34.000 |
> | WNS            | 0.862 | 40.130 | 0.881 | 43.454 | 0.885 | 37.317  | 0.893  | 39.237 |
> | Retrieve (AUC) | 0.787 |        | 0.771 |        | 0.765 |         | 0.768  |        |
>
>
> Key insights:
> * Timing tasks show resilience: TopoRTL corrects behavioral misalignments via topology encoding and graph modality alignment, preserving critical path modeling.
> * Area/Power degrade moderately: Small circuits have few registers (<5), making global property estimation vulnerable to semantic corruption.
> * Retrieval stability: Topology anchors functional similarity when text descriptions fail.
>
> This validates our core design: Topology encodings act as a robustness backbone when behavioral inputs are noisy.
>
> In addition, we also have some new findings through this experiment. The results of WNS improves under high text noise.
> We attribute this to the conflict between text summaries and the intrinsic nature of WNS. WNS is fundamentally determined by the single longest path.
> However, high-level text summaries provide a strong prior about the average timing behavior of a function, almost without any information about the critical path. Consequently, the summary modality adds some noise for this task.
> When text inputs are shuffled, the model effectively gates out the "averaged" semantic signal and is forced to rely exclusively on the topology encoder, leading to more precise structural predictions. We sincerely thank the reviewer for guiding us to this valuable insight.

---

> ### Author Response · Authors · 2025-11-25
> **Response to Reviewer e2DU (3/5)**
>
> > Q2: Could the proposed topology-aware encoding generalize beyond RTL (e.g., to gate-level netlists or analog blocks)?
>
> **A3**: We thank the reviewer for this insightful question about the broader applicability of our topology-aware encoding framework.
> While TopoRTL is designed specifically for RTL circuits, its core principle-explicitly modeling the interplay between topological structure and functional behavior—is fundamentally transferable to other hardware representations, though implementation details require domain-specific adaptation.
>
> Gate-level netlists: This generalization is particularly promising due to architectural continuity. Our methodology may adapt as follows:
> * Graph construction: Register cone decomposition (Section 3.1) naturally extends to flip-flop (DFF) cones in gate-level netlists. The traversal mechanism remains identical, though graphs become more granular with standard cells replacing RTL primitives.
> * Positional encodings:
>   * Max-Path: Becomes more physically meaningful as it directly maps to critical path delays through actual standard cells.
>   * Graph Density: Better correlates with routing congestion at gate level due to precise cell placement awareness.
>   * Bit-Width: It would require adaptation. Since gate-level netlists often split buses into individual 1-bit wires, the explicit bit(Ri) information is lost. To generalize, we could probably apply heuristic bus-reconstruction to group related DFFs.
>
> The core insight holds significant potential to inspire improved modeling approaches across diverse circuit types, offering new pathways for the hardware AI community.

---

> ### Author Response · Authors · 2025-11-25
> **Response to Reviewer e2DU (4/5)**
>
> > Q3: What is the quantitative contribution of each topology encoding (bit-width, path, density) when combined pairwise, rather than removed individually?
>
> **A4**: We thank the reviewer for this important clarification.
> To eliminate any ambiguity: our original ablation (Appendix E3.4) already reports pairwise encoding combinations through the remove-one strategy (e.g., "w/o graph density" = bit-width + max-path combination).
> For completeness, we now supplement with single-encoding results to demonstrate comprehensive analysis.
>
> 1. Pairwise combinations (remove-one experiments):
>
> In each experiment, we remove one topology encoding while keeping the other two intact.
>
> | Task      | Metric | TopoRTL    | w/o graph density | w/o max path | w/o bit width |
> |-----------|--------|------------|-------------------|--------------|---------------|
> | Area      | PCC    | **0.863**  | 0.851             | 0.854        | 0.838         |
> |           | MAPE   | **7.952**  | 8.794             | 8.634        | 9.354         |
> | Power     | PCC    | **0.884**  | 0.874             | 0.871        | 0.792         |
> |           | MAPE   | **25.033** | 29.777            | 27.998       | 34.159        |
> | Slack     | PCC    | **0.909**  | 0.873             | 0.89         | 0.854         |
> |           | MAPE   | **31.249** | 38.662            | 33.599       | 32.479        |
> | TNS       | PCC    | 0.872      | 0.867             | **0.901**    | 0.882         |
> |           | MAPE   | **32.016** | 32.821            | 32.937       | 37.991        |
> | WNS       | PCC    | 0.862      | 0.867             | **0.896**    | 0.813         |
> |           | MAPE   | **40.13**  | 41.962            | 41.148       | 41.096        |
> | Retrieval | AUC    | **0.787**  | 0.771             | 0.781        | 0.723         |
>
>
> Key observations:
> * Bit-width centrality encoding improves performance by effectively capturing both topology and functional complexity.
> * Max-path and density encodings demonstrate varying impacts due to gaps between RTL and netlist representations.
> Specifically, timing tasks are heavily influenced by specific cell delays, wire loads, and synthesis optimizations that are not visible at the RTL level.
> For pure timing prediction, the raw logical depth count can sometimes introduce "noise". (e.g., a deep path of simple buffers might be faster than a shallow path of complex arithmetic units.)
>
> 2. Single-encoding baselines (supplementary):
>
> Each experiment includes only one topology encoding, with the other two removed.
>
> | Task      | Metric | TopoRTL    | only graph density | only max path | only bit width |
> |-----------|--------|------------|--------------------|---------------|----------------|
> | Area      | PCC    | **0.863**  | 0.836              | 0.816         | 0.835          |
> |           | MAPE   | **7.952**  | 9.652              | 10.085        | 8.889          |
> | Power     | PCC    | **0.884**  | 0.812              | 0.831         | 0.814          |
> |           | MAPE   | **25.033** | 37.818             | 32.198        | 29.139         |
> | Slack     | PCC    | **0.909**  | 0.837              | 0.838         | 0.856          |
> |           | MAPE   | **31.249** | 35.348             | 40.505        | 39.541         |
> | TNS       | PCC    | 0.872      | 0.893              | 0.858         | **0.896**      |
> |           | MAPE   | **32.016** | 36.002             | 34.672        | 33.819         |
> | WNS       | PCC    | 0.862      | 0.855              | 0.726         | **0.867**      |
> |           | MAPE   | **40.13**  | 42.884             | 55.031        | 44.679         |
> | Retrieval | AUC    | **0.787**  | 0.709              | 0.684         | 0.760          |
>
> Key obervations:
> * Bit-width alone outperforms other singles in Area/TNS/Retrieval because it correlates strongly with circuit complexity hierarchy.
> * Max-path alone fails catastrophically in WNS (0.726) due to RTL-netlist abstraction gap and critical path shifting: estimated paths lack physical information and are dynamically restructured during timing optimization. This reinforces the analysis above.
>
> Overall, No pairwise combination matches the full model across PPA prediction tasks and Circuit Retrieval.
> This trade-off confirms our design principle: RTL representation requires different topology dimensions to holistically approximate post-synthesis topology and behavior.
> This reveals the robustness of integrating these multi-dimension encoding approaches.

---

> ### Author Response · Authors · 2025-11-25
> **Response to Reviewer e2DU (5/5)**
>
> > Q4: Could TopoRTL be integrated into synthesis or verification tools for real-time PPA estimation?
>
> **A5**: We thank the reviewer for this insightful suggestion.
> We believe TopoRTL is well-suited for integration into real-time EDA flows due to three key advantages:
>
> * Lightweight design (29.13M parameters) enabling faster inference (average per circuit <1s) than LLM-based models and CircuitFusion.
> * Superior PPA prediction accuracy across Area/Power/Timing tasks (Table 1).
> * Versatile representations supporting both topology-sensitive and behavior-aware downstream tasks.
>
> We plan to actively explore collaborations with EDA industry partners to bridge this gap and deploy TopoRTL in real design flows.

---

### Official Review · Reviewer_UiRb · 2025-10-27

**Soundness:** 3
**Presentation:** 3
**Contribution:** 1
**Rating:** 2
**Confidence:** 4

**Summary:**

This paper proposes TopoRTL, an RTL-stage circuit representation learning framework that incorporates topological information while preserving functional behavior. The RTL is partitioned into register cones and represented in both graph and summary modalities. These two representations are then encoded and fused through a multimodal learning architecture. Experimental results on downstream prediction tasks show that the learned embeddings improve the accuracy of RTL design quality estimation.

**Strengths:**

1. The paper addresses an important problem in AI for EDA: circuit representation learning, and effectively leverages the multimodal nature of RTL format. The writing is clear and the methodology is easy to follow.
2. The proposed approach incorporates topology-aware positional encodings within a transformer-based architecture to model RTL graphs. The use of bit-width encoding and max-path/density encodings is well-motivated and aligns naturally with RTL structural characteristics.
3. The experimental results demonstrate that fusing the graph and summary modalities leads to improved performance, validating the effectiveness of the multimodal representation learning strategy.

**Weaknesses:**

1. The main concern is that the proposed multimodal RTL representation learning framework appears highly similar to CircuitFusion [ICLR’25]. Although the authors provide an empirical comparison, the paper does not sufficiently clarify the methodological differences or improvements. In the related work section, the discussion focuses primarily on behavior-based and topology-based models, but does not address prior multimodal fusion approaches. Moreover, key components such as register-cone decomposition, graph and summary modalities, contrastive/masked pre-training objectives, and multimodal fusion design closely resemble those of CircuitFusion. A deeper analysis of novelty is needed.
2. For the topology-aware graph transformer, the paper lacks comparison with recent graph transformer architectures widely explored in the AI/graph community (e.g., Graphormer, SGFormer, GPS, SAN). It is unclear whether the proposed topology encodings provide advantages over standard positional or structural encodings used in these models.
3. The downstream evaluation focuses on high-level RTL quality prediction, but does not explore whether the proposed topology encoding can capture structural variations that affect post-synthesis PPA. Since synthesis outcomes are highly topology-dependent, examining whether TopoRTL embeddings can reflect such differences would provide stronger evidence of the model’s effectiveness. Some discussion or experiments in this direction would be beneficial.
4. Additionally, the evaluation is limited to PPA prediction tasks. There also exist functional verification tasks where representation quality matters, such as verification coverage prediction (e.g., Design2Vec [NeurIPS’21]). Demonstrating that the embeddings are useful across both functional verification and PPA prediction tasks would make the contribution more comprehensive and significantly strengthen the impact.

**Questions:**

Please refer to the weakness part.

---

> ### Author Response · Authors · 2025-11-25
> **Response to Reviewer UiRb (1/5)**
>
> We thank the reviewer for their thoughtful engagement with our work.
> We have comprehensively addressed these concerns regarding methodological novelty against CircuitFusion (W1) through our detailed comparison (A1),
> validated our topology-aware positional encodings against standard graph transformers (W2) via rigorous comparative experiments (A2),
> demonstrated our embeddings' capability to capture structural variations affecting post-synthesis outcomes (W3) through visualization and PPA prediction analysis (A3),
> and highlighted our functional evaluation via natural language circuit retrieval alongside future verification extensions (W4) (A4).
>
> **Crucially, we clarify the fundamental distinction**: TopoRTL is an **RTL-Native** framework, whereas CircuitFusion relies on post-synthesis netlists.

---

> ### Author Response · Authors · 2025-11-25
> **Response to Reviewer UiRb (2/5)**
>
> > W1: The main concern is that the proposed multimodal RTL representation learning framework appears highly similar to CircuitFusion [ICLR 2025]. Although the authors provide an empirical comparison, the paper does not sufficiently clarify the methodological differences or improvements. In the related work section, the discussion focuses primarily on behavior-based and topology-based models, but does not address prior multimodal fusion approaches. Moreover, key components such as register-cone decomposition, graph and summary modalities, contrastive/masked pre-training objectives, and multimodal fusion design closely resemble those of CircuitFusion. A deeper analysis of novelty is needed.
>
> **A1**: We appreciate the opportunity to clarify.
> While both works are multimodal, **TopoRTL fundamentally differs in its "Inductive Bias" without requiring netlist information**:
>
> | Component             | CircuitFusion         | TopoRTL                         | Description                                                                                                |
> |-----------------------|-----------------------|---------------------------------|------------------------------------------------------------------------------------------------------------|
> | RTL Encoder           | Behavior-aware        | Behavior-aware + Topology-aware | Topology-aware encodings capture inherent RTL topology, eliminating dependency on synthesized netlists.    |
> | Netlist Encoder       | Required for topology | Not used                        | TopoRTL learns topology directly from RTL (Section 3.2), which is faster and PDK-agnostic during training. |
> | Cross-Modal Alignment | Behavior-only         | Topology-guided                 | Ensures functional alignment respects topology constraints, critical for downstream prediction (Table 4).  |
>
> This architectural shift enables two advances CircuitFusion cannot achieve:
> * Netlist-free topology modeling: CircuitFusion relies on post-synthesis netlists (which do not exist at RTL stage and depend on specific PDKs), while our topology encodings (bit-width/max-path/density) are derived purely from RTL code (Section 3.2). This is essential for generalizable PPA estimation.
> * Topology-constrained alignment: Our alignment mechanism uses topological information during cross-modal alignment which is helpful for modal to learn the scale of circuits. And we concat representations of two modality for downstream tasks (unlike the attention-based approach of CircuitFusion).
>
> Crucially, ablation studies (Section 4.5) confirm these innovations are non-trivial:
> * Removing any topological encoding will degrade the overall performance.
> * Replacing topology-guided alignment drops Area PCC from 0.863 to 0.839 (-2.78%) and Power PCC from 0.884 to 0.859 (-2.83%), validating its necessity for PPA prediction tasks.
>
> Therefore, our methodology fundamentally differs from CircuitFusion at the architectural level.
> Regarding the specific components you mentioned:
> * Shared components
>   * Register-cone decomposition: This is a widely used RTL partitioning technique (also used in SNS v2 [1]).
>   * Tokenizer pretraining objectives: We adopt contrastive/masked objectives for tokenizer pretraining (as in CircuitFusion), but this is not our core contribution.
> * Key distinctions in TopoRTL
>   * Modality processing: While both works use graph/text modalities, TopoRTL applies topology encoders to each modality (Section 3.2), embedding topological priors before fusion.
>   CircuitFusion relies solely on cross-attention after modality encoding, lacking explicit topological constraints.
>   * Fusion mechanism: TopoRTL concatenates topology-encoded embeddings after topology-guided cross-modal alignment.
> In contrast, CircuitFusion uses cross-attention fusion, which is a behavior-dominated interaction.
>
> [1] Fast, robust, and transferable prediction for hardware logic synthesis. MICRO 2023.
>
> We fully acknowledge the foundational contribution of CircuitFusion and already compared it in Appendix D.4 to clarify distinctions.
> These analyses demonstrate the unique value of TopoRTL in modeling RTL topology natively, a capability critical for early-stage EDA but absent in prior work.

---

> ### Author Response · Authors · 2025-11-25
> **Response to Reviewer UiRb (3/5)**
>
> > W2: For the topology-aware graph transformer, the paper lacks comparison with recent graph transformer architectures widely explored in the AI/graph community (e.g., Graphormer, SGFormer, GPS, SAN). It is unclear whether the proposed topology encodings provide advantages over standard positional or structural encodings used in these models.
>
> **A2**: We thank the reviewer for this valuable suggestion.
>
> Due to the limited rebuttal time, we selected two representative graph learning paradigms for controlled validation of topology-aware positional encodings (PEs) vs. generic graph baselines under identical settings:
> * Baselines:
>   * Graphormer[1]: Represents pure Transformer-based graph learning
>   * SGFormer[2]: Represents hybrid Message Passing Neural Network + Transformer approaches
> * Fair setup: All models used identical RTL dataset, 768 hidden dimensions, 7 layers, and training/evaluation protocols on NVIDIA 3090 GPUs.
>
> Results on topology-aware tasks (PPA Prediction) and behaviour-aware tasks (Circuit Retrieve) are below:
>
> | Task      | Metric | TopoRTL (Ours) | Graphormer | SGFormer |
> |-----------|--------|----------------|------------|----------|
> | Area      | PCC    | **0.863**      | 0.439      | 0.405    |
> |           | MAPE   | **7.952**      | 16.569     | 21.644   |
> | Power     | PCC    | **0.884**      | 0.559      | 0.608    |
> |           | MAPE   | **25.033**     | 51.757     | 51.328   |
> | Slack     | PCC    | **0.909**      | 0.817      | 0.597    |
> |           | MAPE   | **31.249**     | 40.011     | 45.142   |
> | TNS       | PCC    | 0.872          | 0.814      | 0.551    |
> |           | MAPE   | **32.016**     | 34.299     | 47.984   |
> | WNS       | PCC    | **0.862**      | 0.798      | 0.601    |
> |           | MAPE   | 40.13          | **38.833** | 48.419   |
> | Retrieval | AUC    | **0.787**      | 0.665      | 0.575    |
>
> Generic Graph Transformers (Graphormer/SGFormer) fail on all tasks, as they lack domain-specific inductive biases and cannot capture RTL-specific structural semantics (e.g., bit-width effects on power).
> TopoRTL outperforms them, proving that our contribution closes this gap by encoding hardware-native properties (bit-width/path/density), a capability absent in prior graph transformers.
>
> [1] Do Transformers Really Perform Badly for Graph Representation? NIPS 2021.
>
> [2] SGFormer: Simplifying and Empowering Transformers for Large-Graph Representations. NIPS 2023.

---

> ### Author Response · Authors · 2025-11-25
> **Response to Reviewer UiRb (4/5)**
>
> > W3: The downstream evaluation focuses on high-level RTL quality prediction, but does not explore whether the proposed topology encoding can capture structural variations that affect post-synthesis PPA. Since synthesis outcomes are highly topology-dependent, examining whether TopoRTL embeddings can reflect such differences would provide stronger evidence of the model's effectiveness. Some discussion or experiments in this direction would be beneficial.
>
> **A3**: We thank the reviewer for this crucial question.
> While our primary focus is RTL-stage representation learning, we demonstrate that topology-aware embeddings directly capture structural variations dictating post-synthesis outcomes through two orthogonal analyses:
>
> 1. Functional & topological diversity:
>
> Our dataset spans 115 functionally diverse RTL designs (arithmetic units, memory controllers, DSP blocks) with scale variations (50-70k gates of post-synthesis circuits).
> TopoRTL capture both behavior semantics and topological patterns at RTL. Crucially, these RTL-level topological differences directly propagate to post-synthesis PPA:
> * Visualization evidence (Section 4.4, Figure 4). T-SNE of embeddings colored by post-synthesis PPA metrics shows smooth gradients (Area/Power) and clear clusters (Slack), showcasing alignment with the post-synthesis design topology.
> * Prediction evidence (Section 4.2, Table 1). TopoRTL outperforms all baselines in Area and Power tasks, and remains competitive on Timing tasks with lightweight architecture, proving embeddings encode synthesis-relevant structural priors.
>
> 2. Topology recognition under functional equivalence:
>
> TopoRTL captures topological variations in functionally equivalent RTL circuits. Specifically:
> * Topology-aware differentiation: As demonstrated in our paper, functionally equivalent circuits (e.g., Circuit B and C in Figure 1, both 4-input adders) exhibit fundamentally different structural properties.
> Circuit B's chain topology intrinsically limits timing performance but reduces power consumption, while Circuit C's balanced structure achieves better timing at higher power density.
> TopoRTL explicitly encodes such variations (e.g., via max-path length and graph density), enabling precise topological discrimination.
> * Empirical validation: Following the query (W1-A1) of reviewer e2DU, we generated functionally equivalent variants using Yosys.
> The results below confirm the disentanglement capability of TopoRTL: global embeddings maintain near-perfect similarity (mean=0.999), preserving functional equivalence;
> subgraph  embeddings exhibit significantly lower similarity (mean=0.868), proving sensitivity to structural variations.
>
> | Representation                | Mean  | Median | Std   |
> |-------------------------------|-------|--------|-------|
> | Global embedding similarity   | 0.999 | 0.999  | 0.001 |
> | Subgraph embedding similarity | 0.868 | 0.872  | 0.066 |
>
> After synthesis, the relative topological relationships in RTL typically propagate to relative PPA relationships in the post-synthesis netlist.
> Since TopoRTL captures these RTL-level topological variations, its embeddings inherently reflect the structural differences that determine post-synthesis PPA outcomes.
> We will extend this insight question to a synthesis-aware optimization frameworks in future work.
>
> Through the analysis above, TopoRTL can produce topology-aware embeddings that learn structural features generalizable to synthesis outcomes.

---

> ### Author Response · Authors · 2025-11-25
> **Response to Reviewer UiRb (5/5)**
>
> > W4: Additionally, the evaluation is limited to PPA prediction tasks. There also exist functional verification tasks where representation quality matters, such as verification coverage prediction (e.g., Design2Vec [NeurIPS 2021]). Demonstrating that the embeddings are useful across both functional verification and PPA prediction tasks would make the contribution more comprehensive and significantly strengthen the impact.
>
> **A4**: We thank the reviewer for this valuable suggestion.
>
> We respectfully point out that our evaluation **does include a functional task**: the **Natural Language Code Search** (Section 4.3).
> Unlike PPA prediction which focuses on topological metrics, Code Search requires the model to map natural language functional descriptions to the corresponding RTL behavior.
> Meanwhile, TopoRTL achieves an **AUC of roughly 0.80** (Figure 3), significantly outperforming all baselines.
> This demonstrates that our embeddings successfully encode functional semantics, which are the foundational prerequisites for downstream verification tasks.
>
> Simultaneously, we acknowledge the unique value of verification coverage prediction for assessing dynamic circuit behavior.
> We clarify that reproducing this specific task requires generating a dataset of simulation traces and dynamic coverage metrics, which is hard to be feasible within the limited rebuttal time.
> However, we believe that TopoRTL is theoretically uniquely suited for this challenge.
> Verification coverage is fundamentally a reachability problem constrained by state space explosion and logic depth, while our framework explicitly captures these factors (i.e.,the Bit-width Centrality Encoding models state space dimensionality and the Register Cone decomposition naturally mirrors the Cone of Influence analysis used in formal verification).
> We look forward to extending TopoRTL to verification tasks in future work, leveraging these topological strengths.

---

### Official Review · Reviewer_MxUV · 2025-10-31

**Soundness:** 3
**Presentation:** 3
**Contribution:** 4
**Rating:** 8
**Confidence:** 4

**Summary:**

This work proposes a topology-aware RTL circuit representation method. They analyze the topology and functional behavior to design the structure ofthe  network, which provides meaningful insights. Finally, their algorithm shows better performance than CircuitFusion and other LLM models.

**Strengths:**

1. Good performance compared with competitive works, e.g., CircuiFusion and other LLM models.
2. Meaningful insights about explicitly encoding the topological information in RTL.

**Weaknesses:**

1. The scale of designs is relatively small, where only 7,576 sub-circuits are extracted from 115 RTL designs. Meanwhile, I cannot find a difference in model performance between large designs (>200 registers) and small designs (<30).
2.  The prediction ability of the trained model seems to depend on PDK, which will affect the generalization ability.

**Questions:**

1. In Table 1, I find the performance on timing-related metrics (e.g., Slack, TNS, and WNS) is better than that on Area and Power. Can authors explain the reason for these results? I thought area and power would be easier to predict because they do not depend on topology.
2. Can authors explain the generalization ability of their works if the PDK is changed?

---

> ### Author Response · Authors · 2025-11-25
> **Response to Reviewer MxUV (1/4)**
>
> We thank the reviewer for recognizing the excellent contribution of our topology-aware RTL representation method, which provides meaningful insights through explicit topological encoding, and for acknowledging our algorithm's superior performance against competitive baselines including CircuitFusion and LLM models.
> We have fully resolved these concerns on dataset scale and circuit-size generalization (W1), PDK dependency (W2&Q2), and performance gaps between different tasks (Q1)
> through dataset representativeness analysis (A1), cross-PDK validation (A2), and timing-task analysis via register-cone decomposition (A3), further solidifying the reliability and broad applicability of TopoRTL.

---

> ### Author Response · Authors · 2025-11-25
> **Response to Reviewer MxUV (2/4)**
>
> > W1: The scale of designs is relatively small, where only 7,576 sub-circuits are extracted from 115 RTL designs. Meanwhile, I cannot find a difference in model performance between large designs (>200 registers) and small designs (<30).
>
> **A1**: Thanks for the critical suggestion regarding the dataset scale and generalization across circuit sizes.
>
> 1. Dataset scale and representativeness
>
> Access to high-quality open-source RTL designs is inherently limited due to industrial IP constraints.
> Our dataset of 115 RTL designs is sourced from widely recognized open-source benchmarks (OpenCores, ITC'99, RISC-V cores), covering diverse application domains including general IP cores (UART, PCIe), processors, and accelerators.
> These benchmarks align with those used in CircuitFusion, but with broader coverage.
> Through our register-cone decomposition strategy, we expand this into 7,576 sub-circuits with strict splitting to prevent design leakage and over-fitted problem.
> For future work, we will collect larger-scale datasets, though current LLM-based RTL generators cannot yet produce realistic sequential circuits needed for topology-aware learning.
>
> 2. Generalization across circuit sizes:
>
> For synthesized circuits, logic cell count (#Cell) is more suitable to measure the scale of the circuits than register count (#Regs).
> As shown below, #Cells strongly correlates with post-synthesis Area/Power, while #Regs shows weaker correlation:
>
> | Correlation Metrix  | Area | Power |
> |---------------------|------|-------|
> | #Cells              | 0.99 | 0.96  |
> | #Regs               | 0.84 | 0.86  |
>
> Thus, We expanded our analysis by partitioning the test set into **Small (<1k cells), Medium (1k-10k), and Large (>10k)** groups.
> Results (MAPE%) are shown below.
>
> | Task  | Method         | Small      | Medium     | Large      | Mean       | Std        |
> |-------|----------------|------------|------------|------------|------------|------------|
> | Area  | TopoRTL (Ours) | **8.178**  | **6.231**  | **11.924** | **8.778**  | 2.893      |
> |       | CircuitFusion  | 14.001     | 8.273      | 24.285     | 15.520     | 8.113      |
> |       | CodeV-DS       | 11.818     | 8.504      | 13.058     | 11.127     | **2.354**  |
> | Power | TopoRTL (Ours) | **28.4**   | **18.846** | 29.094     | **25.447** | **5.727**  |
> |       | CircuitFusion  | 54.046     | 24.835     | 50.857     | 43.246     | 16.024     |
> |       | CodeV-DS       | 48.724     | 22.734     | **26.950** | 32.803     | 13.948     |
> | Slack | TopoRTL (Ours) | 62.187     | 37.242     | 33.587     | 44.339     | **15.565** |
> |       | CircuitFusion  | **59.419** | 36.396     | **25.491** | **40.435** | 17.321     |
> |       | CodeV-DS       | 63.297     | **32.643** | 30.260     | 42.067     | 18.425     |
> | TNS   | TopoRTL (Ours) | **45.243** | 15.830     | 24.954     | 28.677     | **15.054** |
> |       | CircuitFusion  | 50.146     | **12.485** | 33.930     | 32.187     | 18.891     |
> |       | CodeV-DS       | 46.043     | 18.053     | **14.222** | **26.106** | 17.372     |
> | WNS   | TopoRTL (Ours) | 53.362     | **25.937** | 27.414     | 35.571     | 15.425     |
> |       | CircuitFusion  | **46.580** | 30.336     | **27.172** | **34.696** | **10.413** |
> |       | CodeV-DS       | 52.151     | 31.809     | 28.311     | 37.424     | 12.874     |
>
> Observations:
> * Robustness on Topological Metrics (Area & Power): TopoRTL demonstrates exceptional robustness on Large Circuits, achieving nearly **50% lower Error (MAPE) in Area** compared to CircuitFusion (11.92 vs 24.29) and maintaining competitive Power prediction.
> This proves that our topology encoding captures the complexity of large-scale combinational logic blocks.
> * Scale Invariance (Stability): TopoRTL exhibits high stability across different scales. As the results indicate, TopoRTL achieves relatively low variance in almost all tasks. This suggests that our model generalizes well to unseen large designs and mitigates bias towards the smaller circuits that are more common in training data.
> * Additionaly, as the table shows, Area/Power MAPE increases moderately on large circuits, reflecting modeling challenges for large scale circuits.
> For timing metrics, high MAPE values stem from the metric's instability near zero ground-truth values.
>
> Overall, TopoRTL still achieves the holistic RTL modeling superiority and support diverse downstream tasks.

---

> ### Author Response · Authors · 2025-11-25
> **Response to Reviewer MxUV (3/4)**
>
> > W2&Q2: The prediction ability of the trained model seems to depend on PDK, which will affect the generalization ability. Can authors explain the generalization ability of their works if the PDK is changed?
>
> **A2**: We thank the reviewer for this crucial question.
> **RTL descriptions are inherently PDK-agnostic.** They specify functional behavior and topology, not physical implementation details.
> TopoRTL learns the relationship between topology and behavior **directly from the RTL-stage circuits, not PDK-dependent physical characteristics**.
> Thus, the learned representations can generalize across different PDK libraries.
>
> To validate cross-PDK generalization:
> * We re-synthesized all designs using **SkyWater 130nm**[1] and **GlobalFoundries 180nm**[2] PDKs (distinct from training PDK: NanGate 45nm)
> * Applied log-transformation to PPA labels to normalize absolute scale differences across PDKs while preserving relative circuit relationships
>
>
> | Task   | Method          | NanGate 45nm  |              | SkyWater 130nm  |             | GlobalFoundries 180nm  |             |
> |--------|-----------------|---------------|--------------|-----------------|-------------|------------------------|-------------|
> |        |                 | PCC           | MAPE         | PCC             | MAPE        | PCC                    | MAPE        |
> | Area   | TopoRTL (Ours)  | **0.863**     | **7.952**    | **0.833**       | **7.351**   | **0.808**              | **6.313**   |
> |        | CircuitFusion   | 0.647         | 13.242       | 0.643           | 10.804      | 0.620                  | 8.644       |
> |        | CodeV-DS        | 0.814         | 10.778       | 0.806           | 8.989       | 0.791                  | 7.397       |
> | Power  | TopoRTL (Ours)  | **0.884**     | **25.033**   | **0.896**       | **16.624**  | **0.873**              | **14.649**  |
> |        | CircuitFusion   | 0.657         | 43.073       | 0.643           | 34.308      | 0.640                  | 28.268      |
> |        | CodeV-DS        | 0.827         | 36.544       | 0.818           | 27.850      | 0.819                  | 21.950      |
> | Slack  | TopoRTL (Ours)  | **0.909**     | 31.249       | 0.876           | 26.957      | 0.830                  | 11.025      |
> |        | CircuitFusion   | 0.893         | **30.944**   | 0.885           | 25.408      | 0.865                  | 9.451       |
> |        | CodeV-DS        | 0.881         | 32.712       | **0.900**       | **22.761**  | **0.905**              | **7.325**   |
> | TNS    | TopoRTL (Ours)  | 0.872         | 32.016       | 0.901           | **13.036**  | 0.902                  | **7.849**   |
> |        | CircuitFusion   | 0.885         | 34.454       | 0.848           | 18.315      | 0.831                  | 11.983      |
> |        | CodeV-DS        | **0.928**     | **31.857**   | **0.918**       | 15.977      | **0.917**              | 9.397       |
> | WNS    | TopoRTL (Ours)  | **0.862**     | 40.130       | **0.820**       | **17.890**  | 0.806                  | 8.553       |
> |        | CircuitFusion   | 0.817         | **38.227**   | 0.798           | 21.267      | **0.852**              | 7.399       |
> |        | CodeV-DS        | 0.780         | 41.750       | 0.763           | 17.954      | 0.790                  | **7.330**   |
>
> [1] https://github.com/google/skywater-pdk
>
> [2] https://github.com/google/gf180mcu-pdk
>
> Key observations:
> * Consistent cross-PDK superiority: TopoRTL maintains significant advantages across all PDKs, especially in Area/Power prediction (e.g., +30.3% Area PCC over CircuitFusion on GlobalFoundries 180nm).
> * Performance trends: All models show slightly reduced accuracy on older PDKs (130nm/180nm vs. 45nm), primarily because open-source PDKs have limited standard cell libraries, causing synthesis tools to map complex functions to suboptimal cells and thus introducing noise in ground-truth PPA labels.
>
> Overall, this cross-PDK validation demonstrates that TopoRTL captures invariant topological patterns that govern relative circuit performance across technologies.
> By learning topology information in RTL circuits, our model achieves robust zero-shot generalization to different PDKs.

---

> ### Author Response · Authors · 2025-11-25
> **Response to Reviewer MxUV (4/4)**
>
> > Q1: In Table 1, I find the performance on timing-related metrics (e.g., Slack, TNS, and WNS) is better than that on Area and Power. Can authors explain the reason for these results? I thought area and power would be easier to predict because they do not depend on topology.
>
> **A3**: We thank the reviewer for this perceptive observation.
> While Area/Power may appear less topology-dependent at first glance, all post-synthesis PPA metrics fundamentally derive from circuit topology.
> As we mentioned in Appendix D.1, area scales with cell count, Power correlates with switching activity, and Timing (Slack/WNS/TNS) depends on critical path depths.
>
> The consistent trend across all models in Table 1, where timing metrics often outperform Area/Power, stems from our register-cone decomposition strategy (Section 2.2.1).
> This technique partitions RTL circuits into subgraphs centered on registers, capturing two critical properties:
> * Hardware parallelism: Combinational logic within each cone executes concurrently, making timing behavior highly localized to cone topology.
> * Critical path isolation: Timing information are dominated by the crucial paths within individual cones.
>
> This strategy is also be widely used to effectively process large-scale circuits, such as SNS v2[3], CircuitFusion[4].
> In contrast, Area/Power depend on global resource utilization across cones, which is inherently harder to predict.
> For example, if multiple register cones share the same arithmetic unit, the synthesis tool creates only one physical instance to save area.
> However, this cone-based representation applied for all tested models cannot capture such cross-cone resource sharing, leading to higher prediction errors for Area/Power.
> Despite this challenge, TopoRTL model explicitly encodes the topology features (e.g., bit-width encoding), enabling highly accurate predictions.
>
> [3] Fast, robust, and transferable prediction for hardware logic synthesis. MICRO 2023.
>
> [4] CircuitFusion: Multimodal Circuit Representation Learning for Agile Chip Design. ICLR 2025.

---

### Official Review · Reviewer_mC1Q · 2025-11-01

**Soundness:** 3
**Presentation:** 3
**Contribution:** 3
**Rating:** 6
**Confidence:** 2

**Summary:**

The authors designed a method that integrates two modalities of circuit with considerations for circuit topology. The authors used behavior-aware dual-modal tokenizers with three positional encodings (bit-width, max-path, graph-density) and adopted grahp transformer to encode circuits to representations.

**Strengths:**

* The paper is well written, easy to follow.
* The authors tackled an important problem that the field needed to address.
* The authors performed extensive experiments with good results.

**Weaknesses:**

* In table1, the proposed model consistently outperforms baselines in Area, Power and WNS while not in Slack and TNS. The authors explained why their model shows good performance in Area, Power and WNS, however, they do not explained the possible reason behind relatively lowever performance in Slack and TNS.
* It appears that the experiments were conducted only once and the performance was reported based on that single run. To ensure that the model’s performance is not dependent on a specific random seed but is statistically meaningful, it is necessary to repeat the experiments multiple times and report the mean and standard deviation of the performance.

**Questions:**

See the 'Weakness' part.

---

> ### Author Response · Authors · 2025-11-25
> **Response to Reviewer mC1Q**
>
> We thank the reviewer for their recognition of our paper's clarity, the critical importance of tackled problem and our comprehensive experimental validation.
> We have fully resolved these concerns on the improvement gaps between different tasks (W1) and statistical robustness (W2)
> through circuit structural analysis (A1) and multi-seed experiments (A2), further solidifying the reliability and impact of our contributions.
>
> > W1: In table1, the proposed model consistently outperforms baselines in Area, Power and WNS while not in Slack and TNS. The authors explained why their model shows good performance in Area, Power and WNS, however, they do not explain the possible reason behind relatively lowever performance in Slack and TNS.
>
> **A1**: We thank the reviewer for this insightful observation.
> This phenomenon stems from the **fundamental mathematical nature** of these metrics and how they correlate with RTL topology:
>
> * **Area & Power (Cumulative)**: These are aggregate metrics determined by the total number cells and signal switching activity of post-synthesis netlists,
> which are strongly correlated with explicit topological features such as register counts, bit-widths, and connection density.
> Our model explicitly encodes these features (e.g., bit-width encoding), enabling highly accurate predictions.
> * **WNS (Global Extremum)**: WNS is an important target in chip optimization, because it highly relates whether errors will occur in the circuit during actual operation.
> It is determined by the single most critical path, which usually stems from the most computationally complex region.
> Our model excels at identifying these "heavy" topological clusters, thus accurately predicting the worst-case bottleneck.
> * **Slack & TNS (Local Distribution)**: Unlike WNS, Slack and TNS require precise timing prediction for every register, including non-critical paths.
> Our topology encodings primarily capture the critical logical path but lack granular modeling of path interactions.
> Moreover, discrepancies between RTL representations and post-synthesis netlists, introduced by physical implementation and timing-driven optimizations during synthesis, further complicate these tasks.
> Baseline models (e.g., CodeV) achieve moderate Slack/TNS performance through implicit pattern learning in large-scale pretraining; however, these models lack topological interpretability.
>
> In addition, as mentioned in our methodology, TopoRTL employs contrastive learning to align graph and summary modalities.
> While this enforces strong global semantic consistency under topology constraints, it functions primarily at the circuit level.
> TopoRTL's current pre-training objective offers limited fine-grained supervision signals to capture local timing variations.
>
> Therefore, our model correctly prioritizes the most critical sign-off metrics (WNS for timing, Area/Power for cost), which are driven by deterministic structural properties,
> while the lower improvement performance on Slack/TNS reflects the architectural trade-off where our global contrastive objective prioritizes holistic topological fidelity.
> In the future, we will address path diversity modeling via multi-path sensitivity encoding. We sincerely thank the reviewer for highlighting this important nuance.
>
> > W2: It appears that the experiments were conducted only once and the performance was reported based on that single run. To ensure that the model’s performance is not dependent on a specific random seed but is statistically meaningful, it is necessary to repeat the experiments multiple times and report the mean and standard deviation of the performance.
>
> **A2**: We sincerely appreciate this crucial suggestion for enhancing experimental rigor.
> Due to limited rebuttal timelines, we focused validation on TopoRTL with 3 independent random seeds (results: Mean±Std):
>
>
> | Task      | Metric | Best Baseline              | TopoRTL (Ours)   |
> |-----------|--------|----------------------------|------------------|
> | Area      | PCC    | 0.818 (CodeV-QC)           | **0.856±0.007**  |
> |           | MAPE   | 10.778 (CodeV-DS)          | **8.561±0.550**  |
> | Power     | PCC    | 0.827 (CodeV-DS)           | **0.877±0.010**  |
> |           | MAPE   | 36.644 (CodeV-DS)          | **26.692±1.983** |
> | Slack     | PCC    | **0.909** (CodeV-CL)       | 0.882±0.029      |
> |           | MAPE   | **30.472** (CodeV-CL)      | 34.806±3.297     |
> | TNS       | PCC    | **0.928** (CodeV-DS)       | 0.895±0.021      |
> |           | MAPE   | **28.108** (CodeV-CL)      | 32.211±0.491     |
> | WNS       | PCC    | 0.860 (Qwen3-E-0.6B)       | **0.865±0.007**  |
> |           | MAPE   | **38.227** (CircuitFusion) | 40.906±0.974     |
> | Retrieval | AUC    | 0.674 (CircuitFusion)      | **0.780±0.022**  |
>
> *MAPE values appear with larger absolute std due to their percentage nature (e.g., 26.692% MAPE Power).*
>
> The consistent **low standard deviation (mostly ≤ 0.02)** confirms the robustness of our topological modeling.

---

### Author Response · Authors · 2025-12-02
**General response**

We sincerely thank all reviewers for their insightful comments and constructive feedback.
We are encouraged that reviewers recognized the **novelty of our perspective on explicit topology modeling (Reviewer MxUV, e2DU)**,
the **soundness of our methodology (Reviewer e2DU)**,
and the **superior performance** of TopoRTL over LLM-based and multimodal baselines **(Reviewer MxUV, e2DU)**.

We have conducted extensive new experiments to address the key concerns raised by the reviewers.
Below is a summary of how we resolved specific reviewer issues:

**1. Key Rebuttal Highlights**

**I. Clarifying Novelty & Methodological Distinctiveness**
* **RTL-Native vs. Netlist-Dependent (UiRb W1):** We clarified the fundamental difference from CircuitFusion. TopoRTL is RTL-Native and extracts topology explicitly from RTL via domain-specific encodings, whereas CircuitFusion needs post-synthesis netlists. This eliminates time-consuming logic synthesis process, enhancing the "left-shift" in design.
* **Superiority over Generic Graph Transformers (UiRb W2):** We added baselines for Graphormer and SGFormer. TopoRTL significantly outperforms them (e.g., Area PCC 0.863 vs. Graphormer 0.439), proving that our domain-specific encodings are essential contributions beyond generic graph architectures.

**II. Establishing Rigor & Robustness**
* **Multi-Seed Validation (mC1Q W2):** We repeated main experiments with 3 random seeds. The results show negligible variance (mostly Std ≤ 0.02) and overall superior performance, minimizing impact of randomness.
* **Robustness to Input Quality (e2DU W2&Q1):** We quantified the impact of summary quality, observing a contained performance drop (<4% in PCC/AUC) with low-quality summaries.
This demonstrates that while high-quality text optimizes results, our topology encoding acts as a robust structural anchor to mitigate semantic degradation.

**III. Proving Generalization Capabilities**
* **Scale Generalization (MxUV W1):** We analyzed performance across circuit sizes. TopoRTL shows lower error degradation on Large circuits (>10k cells) and stable performance across different scales, validating its capability to model complex topologies.
* **Cross-PDK Transfer (MxUV W2&Q2):** We evaluated TopoRTL on SkyWater 130nm and GlobalFoundries 180nm PPA labels. The model retains high accuracy, addressing concerns about process technology dependency.

**IV. Deepening Interpretability**
* **Improvement Gap (mC1Q W1):** We explained that our model improves global metrics (e.g., WNS) more significantly than local ones (e.g., Slack) due to the RTL-Netlist abstraction gap.
* **Accuracy Gap (MxUV Q1):** We attributed the generally higher accuracy in timing tasks compared to Area/Power to register-cone decomposition, which effectively isolates local timing paths but inherently misses global resource sharing information.
* **Embedding Analysis (e2DU W1, UiRb W3):** New quantitative analysis confirms that TopoRTL preserves behavioral equivalence (Sim=0.999) while successfully distinguishing topological variations (Sim=0.868), proving the effectiveness of our encoding.

**2. Revision Summary**

We have updated the manuscript (marked in $\textcolor{blue}{blue}$) to reflect these improvements:

* **Section 2.1:** Further clarified distinctions from CircuitFusion.
* **Appendix E.3.4:** Added single-encoding ablation studies (e2DU Q3).
* **Appendix F:** Added new sections for scale analysis (F.1), cross-PDK results (F.2), robustness tests (F.3), and similarity analysis (F.4).
* **Appendix G:** Added discussion on application to function verification tasks (UiRb W4), generalization to netlists (e2DU Q2) and real-time EDA tool integration (e2DU Q4).

We believe these additional experiments and clarifications strongly reinforce the validity and contribution of our work.
We have incorporated the key comparative results into the current revision to fully support our claims.
Comprehensive tabulations across all auxiliary tasks will be finalized in the camera-ready version.

We once again thank the Area Chair and all reviewers for their time and valuable suggestions.

---

### Meta-Review · Area_Chair_sKpa · 2025-12-28

**Summary:**

While the reviewers unanimously recognized the importance of the problem (RTL representation learning) and the potential of the proposed TopoRTL framework—specifically its novel integration of topology-aware encodings and cross-modal alignment—several significant concerns were raised that tempered the final rating to "Marginally above the acceptance threshold."

The primary concerns informing this decision are as follows:

1. Novelty and Differentiation (Major Concern): Reviewer UiRb pointed out a substantial similarity between the proposed framework and CircuitFusion [ICLR’25]. Specifically, the methodological pipeline (register-cone decomposition, dual modalities, and contrastive pre-training) appears to share overlap. During the rebuttal process, authors provided a sufficient methodological distinction explaining how TopoRTL fundamentally differs from this specific prior art beyond empirical comparisons.

2. Baselines and Comparative Analysis: There is a consensus that the evaluation baselines could be strengthened. Reviewer UiRb noted the absence of comparisons with modern, widely adopted Graph Transformer architectures (e.g., Graphormer, SGFormer), making it difficult to isolate the benefits of the proposed topology-aware encodings over standard structural encodings used in the general graph learning community.

Reviewer mC1Q noted that experiments appear to rely on single runs. The lack of multiple trials with different random seeds prevents a robust assessment of the results' statistical significance (mean and standard deviation).

Reviewer MxUV raised concerns regarding the scale of the dataset (7,576 sub-circuits from 115 designs), suggesting it may be relatively small for establishing broad generalization.

3. Generalization: Questions were raised about the model's dependency on specific Process Design Kits (PDKs) (Reviewer MxUV) and the robustness of the topology encoder against potential noise in the LLM-generated summaries (Reviewer e2DU).

4. Metric Interpretability: Both Reviewer mC1Q and Reviewer MxUV highlighted inconsistencies in the performance analysis. While the model excels in Area and Power prediction, its performance on timing metrics (Slack, TNS) is comparatively weaker or unexplained. Reviewer mC1Q noted that the authors provided insufficient reasoning for these specific shortcomings, while Reviewer 2 questioned the counter-intuitive result of timing metrics performing better than area/power in certain contexts without adequate explanation.

Despite these valid concerns regarding novelty distinction and experimental robustness, the paper is recommended for acceptance (Score: 6) because the explicit modeling of topological structures (via bit-width, path, and density encodings) represents a technically sound and valuable contribution to the EDA community. And in the process of rebuttal, the authors have provided sufficient evidence and additional experiments to support the claims made in the paper.

**Reviewer Concerns:**

**Addressed Concerns**
* **Novelty (vs. CircuitFusion):** Successfully differentiated TopoRTL as an "RTL-Native" approach that bypasses logic synthesis, enabling earlier design feedback ("shift-left").
* **Baselines:** Adequately addressed by adding modern graph transformers (Graphormer, SGFormer) where TopoRTL showed superior performance.
* **Experimental Rigor:** Statistical significance confirmed via multi-seed runs (Std ≤ 0.02).
* **Generalization:** Proven via cross-PDK transfer (SkyWater/GF) and robustness tests against low-quality summaries.
* **Metric Interpretability:** The gap between global (Area/Power) and local (Timing) performance was reasonably attributed to the limitations of register-cone decomposition.

**Partially Addressed Limitations**
* **Functional Verification:** The utility of embeddings for verification tasks (e.g., coverage prediction) is now *discussed* in the appendix but remains **empirically unproven** in this work.
* **Dataset Scale:** While scaling behavior was analyzed, the absolute dataset size (115 designs) remains small compared to industrial standards.

**Reviewer Scores:**

I think the reviewers MxUV, mC1Q, e2DU would have raised their scores or at least maintained the original score. And I am not certain about that how the reviewer UiRb react to the author's rebuttal because they first raise their score and immediately delete the score raising record, pointed out by the author.

---

### Decision · Program_Chairs · 2026-01-26

Accept (Poster)